# An epitaxial graphene platform for zero-energy edge state nanoelectronics

Vladimir S. Prudkovskiy[1,2,3], Yiran Hu[2], Kaimin Zhang[1,11], Yue Hu[2,11], Peixuan Ji[1], Grant Nunn[2], Jian Zhao[1], Chenqian Shi[1], Antonio Tejeda [4,5], David Wander [3], Alessandro De Cecco[3], Clemens B. Winkelmann [3], Yuxuan Jiang[6], Tianhao Zhao[2], Katsunori Wakabayashi [7,8], Zhigang Jiang [2], Lei Ma[1,9] ✉, Claire Berger[2,3,10] & Walt A. de Heer[1,2] ✉

Graphene's original promise to succeed silicon faltered due to pervasive edge disorder in lithographically patterned deposited graphene and the lack of a new electronics paradigm. Here we demonstrate that the annealed edges in conventionally patterned graphene epitaxially grown on a silicon carbide substrate (epigraphene) are stabilized by the substrate and support a protected edge state. The edge state has a mean free path that is greater than 50 microns, 5000 times greater than the bulk states and involves a theoretically unexpected Majorana-like zero-energy non-degenerate quasiparticle that does not produce a Hall voltage. In seamless integrated structures, the edge state forms a zero-energy one-dimensional ballistic network with essentially dissipationless nodes at ribbon–ribbon junctions. Seamless device structures offer a variety of switching possibilities including quantum coherent devices at low temperatures. This makes epigraphene a technologically viable graphene nanoelectronics platform that has the potential to succeed silicon nanoelectronics.

A viable nanoelectronics platform can be defined as a material that can be processed using conventional nanoelectronics technology, as required to produce high density, high performance commercial nanoelectronics. Currently, silicon provides a platform for high performance complementary metal oxide semiconductor (CMOS) nanoelectronics that enables billions of transistors to be patterned on an area of 1 cm² using the most highly developed technologies in the world. However, the CMOS platform is reaching its scaling limits[1]. A viable continuation will require a paradigm shift in electronics, while preserving, as much as possible, current industrial fabrication methods[2]. The latter condition implies that this new platform should

involve a single crystal substrate and a conventionally nanopatternable material. We show here that electronics based on the epigraphene edge state is perfectly suited: electronics grade single crystal SiC is already used in commercial electronics[3] and the edge state charge carrier is a new quasiparticle. In contrast, electronics approaches where chemically produced graphene ribbons are proposed to be interconnected with metal nanowires is based on conventional transistor concepts and still poses daunting technical challenges before they are technologically viable[4,5].

The high electronic mobility and long mean free paths in rolled up graphene sheets (i.e., carbon nanotubes) suggested a graphene

[1]Tianjin International Center for Nanoparticles and Nanosystems (TICNN), Tianjin University, Nankai District 30007, China. [2]School of Physics, Georgia Institute of Technology, Atlanta, GA 30332, USA. [3]Institut Néel, Univ. Grenoble Alpes, CNRS, Grenoble INP, 38042 Grenoble, France. [4]Laboratoire de Physique des Solides, CNRS, Univ. Paris-Sud, 91405 Orsay, France. [5]Synchrotron SOLEIL, L'Orme des Merisiers, Saint-Aubin, 91192 Gif sur Yvette, France. [6]National High Magnetic Field Laboratory, Tallahassee, FL 32310, USA. [7]School of Science and Technology, Kwansei Gakuin University, Gakuen 2-1, Sanda 669-1337, Japan. [8]Center for Spintronics Research Network (CSRN), Osaka University, Toyonaka 560-8531, Japan. [9]State Key Laboratory of Precision Measurement Technology and Instruments, Tianjin University, Nankai District 300072, China. [10]Laboratoire de Recherche International 2958 Georgia Tech-CNRS, 57070 Metz, France. [11]These authors contributed equally: Kaimin Zhang, Yue Hu. ✉e-mail: maleixinjiang@tju.edu.cn; deheer.walt@gmail.com

nanoelectronics platform as a successor to the silicon nanoelectronics platform[1,6–11]. The graphene platform promised to extend the Si miniaturization limit while at the same time introducing a new electronics paradigm by enabling quantum phase coherent devices[6,12,13] as first demonstrated in carbon nanotubes[8,9]. The graphene platform requires graphene nanostructures to be lithographically patterned from two-dimensional graphene sheets to produce the seamlessly interconnected graphene device structures. This cannot be achieved with the alternative bottom-up approach where chemically produced graphene ribbons are interconnected with metal nano wires, which destroys quantum coherence and severely limits performance. While important for flexible electronics, it is not competitive with CMOS technology[4,11]. Despite initial speculations that exfoliated graphene and deposited graphene could be used to realize this graphene nanoelectronics platform[14], pervasive 10 nm scale edge disorder[15–18] dashed hopes of success[19] because an intact edge is required to produce a bandgap in armchair graphene ribbons and the edge state in chiral graphene. Moreover, the edge state is highly desired in graphene interconnected electronics since it is the only transporting state in neutral graphene nanostructures[20–23].

The high quality epigraphene layer that grows epitaxially on silicon carbide (SiC)[12,13,24–27] can be nanopatterned[6,13,28–31]. SiC is a mainstream electronic material[3,32] that is compatible with industrial nanoelectronics processing methods[32–34] that now can produce nanostructures as small as 5 nm[35]. This is relevant because a 5 nm wide graphene armchair ribbon has a predicted band gap greater than 0.2 eV[36,37]. Moreover, SiC is compatible with THz electronics. Chiral graphene ribbons with intact edges are metallic due to a zero-energy edge state (zero-mode) in the gap[21,38,39]. With current technology, large scale graphene networks composed of seamlessly integrated metallic and semiconducting epigraphene nanoribbons can be produced[6,13]. The absence of a bandgap in graphene is often cited as a showstopping problem[19], however already in 2002 Wakabayashi demonstrated[22] that the gapless graphene edge state can also be gated in principle (see also Supplementary Information SI14).

The topologically protected edge state[20–23,38–40] was predicted in 1996 and edge state transport was first observed[30,41] in 2014 up to room temperature in self-assembled 40 nm wide graphene ribbons grown on the sidewalls[29,31] of thermally annealed trenches etched in single crystal silicon carbide wafers[29–31,41–43] (see also SI15). However, sidewall ribbons cannot be seamlessly integrated, and their topology severely limits conventional transport measurements, so that little was known experimentally prior to this work.

To verify the viability of the epigraphene zero-mode based platform we demonstrate seamlessly interconnected charge-neutral ballistic epigraphene ribbon networks using standard nanolithography methods[12,13,25,27] on non-polar silicon carbide substrates. Exceptional edge state transport is observed with unprecedented mean free paths exceeding 50 microns even at room temperature. This is 5000 times greater than the mean free path of the bulk states on the same ribbon, which means that the epigraphene edge state survives the harsh nanolithographic processes and it is not affected by deposited amorphous dielectric coatings. Hence, the epigraphene zero mode platform[6,12] is unique and it enables edge state nanoelectronics.

Theoretically, the graphene edge state[20–23,38–40,44–46], which should not be confused with quantum Hall states[14,47–49] and other magnetically induced edge states[50], is a zero-energy mode of graphene ribbons associated with the edge-localized flatband[20,21,38–40,46]. Its basic electronic structure for zigzag ribbons[20,21], shown in Fig. 1a, generically applies to all chiral ribbons[21,38,39]. We find that the large density of states caused by the flatband (the 0-DoS peak)[21,51] pins the Fermi level at energy $E = 0$ at the edges (Fig. 1b)[51]. Moreover, a Schottky barrier[52] at the edge insulates the edge state from the bulk states resulting in an isolated 1D zero-energy ballistic edge state network with essentially elastic scattering at the ribbon–ribbon junctions. Consequently, heat is

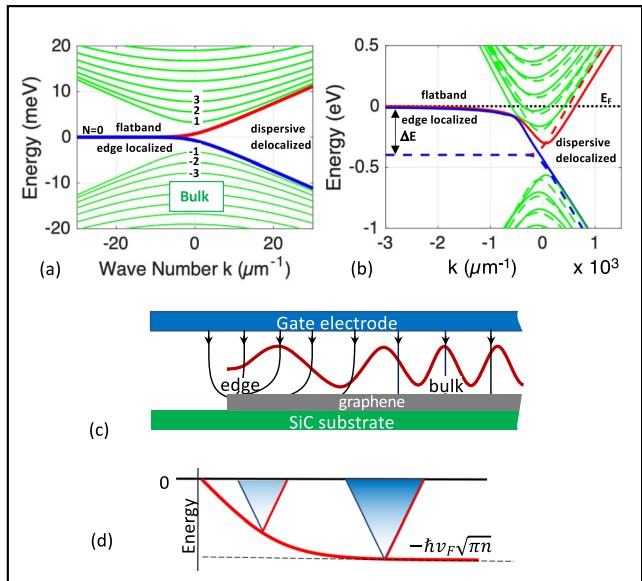

**Fig. 1 | The epigraphene edge state. a** Tight binding band structure of one valley of a 740 nm wide zigzag graphene ribbon that consists of the hyperbolic bulk 1D subbands and the edge state. The edge state is composed of a flat band at E=0 that is localized at the ribbon edge and that merges into two delocalized linear dispersing branches. Original theory predicted that the delocalized branches of the $N = 0$ subband is a protected edge state for energies between the $N = \pm 1$ bulk subbands[40]. **b** Tight binding band structure where the flat band at energy $E = 0$ pins the Fermi level $E_F$ at $E = 0$ at the edge. Charges induced near the edge will be depleted by the flatband to produce a Schottky barrier between the edge and the bulk. The resulting electric fields cause band bending so that the Dirac point will be at $\hbar v_F \sqrt{\pi n}$ below the Fermi level, as schematically depicted in (**d**). **c** Schematically shows the divergence of the Fermi wavelength at the edge.

primarily generated at the metal-ribbon contacts and not at the ribbon-ribbon junctions[53]. We also find that the edge state does not generate a Hall voltage, which suggests that the edge state is an ambipolar single channel state that is half-electron and half-hole in contrast to the theory that predicts that it is either an electron or a hole.

Since the edge state properties presented here defy current theory, they require rigorous experimental evidence. To that end, we first demonstrate that the epigraphene on the non-polar substrates used here is charge neutral graphene. We explain how the silicon carbide substrate stabilizes the edges. We next demonstrate that edge state transport involves a one-dimensional non-degenerate ballistic channel that scatters at graphene junctions. Then we demonstrate that the edge state is pinned at zero energy and that it does not generate a Hall voltage. We conclude with examples of device architectures.

## Results

### 2D nonpolar epigraphene and edge stabilization

The SiC wafers are cut and polished in-house from hexagonal SiC stock to expose nonpolar $(1\bar{1}0n)$, $n \approx 5$, surfaces. As shown in Fig. 2 the thermally grown epigraphene[25] has conventional neutral graphene properties (see also SI12). Scanning tunneling and electron microscopies confirm the graphene lattice structure and its epitaxial alignment with the SiC substrate (Fig. 2a)[12,27,54,55]. This is essential to accurately direct the lithography, however as discussed below, even an annealed edge is generally not straight (Fig. S20) so that the actual edges do not have a well-defined chirality. Raman spectroscopy shows a narrow 2D peak and a small D peak (Fig. 2b)[56]. Angle resolved photoelectron spectroscopy (Fig. 2c) shows the characteristic Dirac cones with their apex at $E = 0$, corresponding to neutral graphene with a Fermi velocity $v_F = 1.06 \times 10^6$ m/s[57–59]. Infrared magneto-spectroscopy (Fig. 2d) verifies the graphene electronic dispersion: $E(B) = v_F \sqrt{2|N_{LL}|e\hbar B}$, where

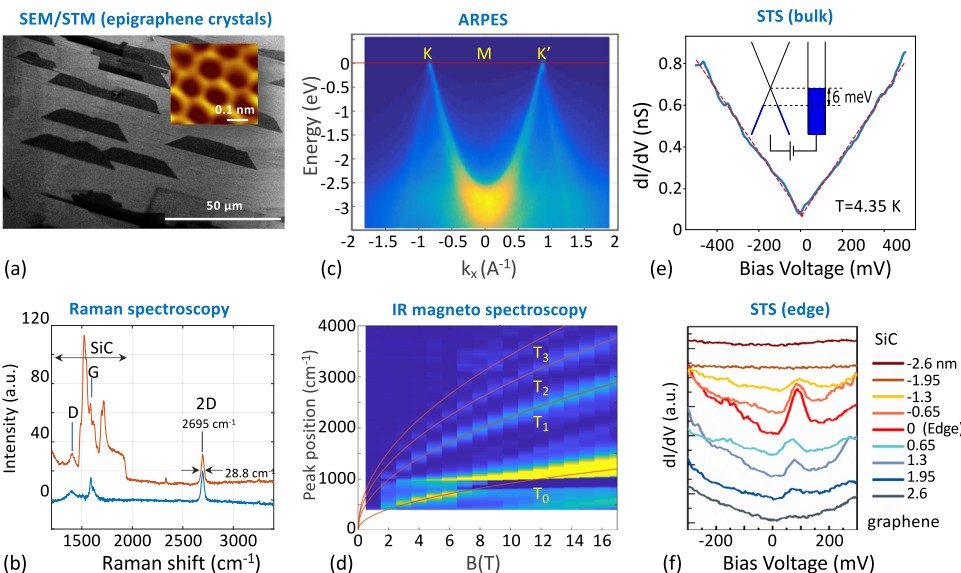

**Fig. 2 | Neutral epigraphene characterization. a** SEM micrograph of trapezoidal graphene islands that form early in the growth and ultimately coalesce to produce a uniform graphene layer. (Inset) STM image of the epigraphene showing the characteristic hexagonal lattice of graphene at $T = 12.5$ K. **b** Raman spectroscopy. Measured spectrum (red) and SiC subtracted spectrum (blue). The 2D peak is typical of a graphene monolayer. **c** ARPES (beam energy = 200 eV, $E_F = 197.4$ eV) taken at room temperature along K-M-K' showing characteristic graphene Dirac cones with $v_F = 1.06 \times 10^6$ m/s, with an apex at $E = 0$ confirming charge neutrality and no detectable anisotropy. **d** Infrared magneto-spectroscopy. The transitions follow the expected characteristic graphene $\sqrt{B}$ dispersion (indicated by the red lines) confirming its monolayer character. **e** Typical scanning tunneling spectrum ($T = 4.4$ K, $I_{set} = 400$ pA at $V_{bias} = 500$ mV), showing the characteristic graphene density of states. A linear fit (dashed lines) indicates a doping level $|E_F - E_D| < 6$ meV, showing that the graphene is charge neutral. **f** STS image at a graphene island edge ($T = 12.5$ K, $I_{set} = 250$ pA at $V_{bias} = 2$ V) taken at various distances from the edge from SiC to inside the ribbon with a lateral resolution of about 2 nm (traces are displaced vertically for clarity). Note the 0-DoS peak at the edge, similar to that observed in sidewall ribbons[43].

$N_{LL}$ is the Landau level index. Scanning tunneling spectroscopy (STS) (Fig. 2e) verifies that the Fermi level $E_F$ is close to $E_F = 0$ and STS measured at a series of locations across the edge of a graphene crystal shows the 0-DoS peak at the edge (Fig. 2f), similar to that observed in ref. 60.

We find that only thermally annealed epigraphene structures have an edge state (See SI16)[61]. In Method 2 (see SI3), the epigraphene is patterned by reactive oxygen ion etching (RIE) followed by annealing in vacuum at 1200 °C. Method 1 (SI3) involves inductively coupled plasma (ICP) etching[62], that cuts through the graphene and 16 nm into the SiC substrate via an e-beam patterned resist mask. Like plasma cutting and welding, energetic ion collisions produce extremely high local temperatures ($T > 5000$ K)[62,63] that cause carbon and silicon to evaporate. During cooling, C–C and Si–C bands are formed[63], fusing the graphene edges to the silicon carbide thereby chemically and mechanically stabilizing the edges[30,64–66]. Cross-sectional electron microscopy indicates that thermally annealed sidewall ribbons are bonded to the substrate via acene (i.e., zigzag-like) edge atoms[67,68] that have a single unoccupied state at $E = 0$[69]. In graphene ribbons these edge atoms cause the 0-DoS peak that manifests as the $E = 0$ flatband[21,39,67,68], which is essential for the graphene edge state[38,39]. The fact that edge state is also seen in meandering ribbons (SI15)[30,41] indicates that nominally zigzag (ZZ) and armchair (AC) edges are acene bonded in general. Here we use the ZZ and AC designation to distinguish these two directions and not to indicate the actual graphene edge morphology, which for annealed edges is determined by their stability that involves the bonding to the substrate, mostly likely favoring acene bonded edges (See SI17). The Si–C bond is slightly polar[63,70] so that the edges will be very slightly n or p-doped depending on their bonding to the SiC substrate.

Three Hall bars labeled S1–S3 were prepared for this study of which S1 and S2 are discussed in detail. They have similar geometries (Figs. 3b, e and SI3) but with production and dimension variations. The long arm of Sample S1 prepared with ICP etching (Method 1, Fig. S4) is along the AC direction and Sample S2, prepared with Method 2 (Fig. S5), is along the ZZ direction.

Transport measurements were performed at temperatures from $T = 2$ to 300 K and magnetic fields $B$ up to 9 T. As shown in Fig. 3b, e and SI3, the Hall bar structures are composed of graphene ribbon segments, **A** through **J**. Adjacent segments are seamlessly connected by edgeless, square graphene junctions. The gate fully covers only segments **B** and **C**. While bulk epigraphene is intrinsically charge neutral (Fig. 2), processing charges the graphene, with a charge density $n_0$: $n_0 \approx 10^{12}$ cm$^{-2}$ in S1 and $2 \times 10^{11}$ cm$^{-2}$ in S2, so that the charge density of ungated segment sections is $n_0$. We normalize $V_G$ so that $V_G = 0$ corresponds to the charge neutrality point (CNP), i.e., where the conductance has a minimum and the Hall voltage switches sign (see below). Resistances $R_{ij,kl}$ are determined with current $I_0$ flowing from contact $i$ to $j$, and voltages measured between contacts $k$ and $l$. Conductances $G_{ij,kl}$ are defined by $G_{ij,kl} = 1/R_{ij,kl}$.

### Characterization of the single channel edge state

The following systematic study (1) unambiguously identifies ballistic transport of the edge state from scaling; (2) reveals differences in AC and ZZ edges; (3) shows the insensitivity of the edge state to the gate voltage; (4) demonstrates isotropic scattering at the junctions; (5) shows the reduction of backscattering in a magnetic field. (6) Edgeless Corbino ring[71] measurements (SI2) confirm that the observed properties are due to the edge state.

The conductance of a diffusive graphene ribbon of length $L$ and width $W$ is $G = \sigma W/L$. The conductivity $\sigma = GL/W = ne\mu$ ($n$ is the charge density and $\mu$ the mobility) is a material property of diffusive conductors that does not depend on size or shape and therefore imposes strict scaling rules. In contrast, in a single channel ballistic conductor[53] $G = G_0(1 + L/\lambda)^{-1}$ so that the conductance is not inversely proportional to the length. Here, $G_0 = e^2/h$, $e$ is the electronic charge, $h$ is Planck's constant, and $\lambda$ is the mean free path.

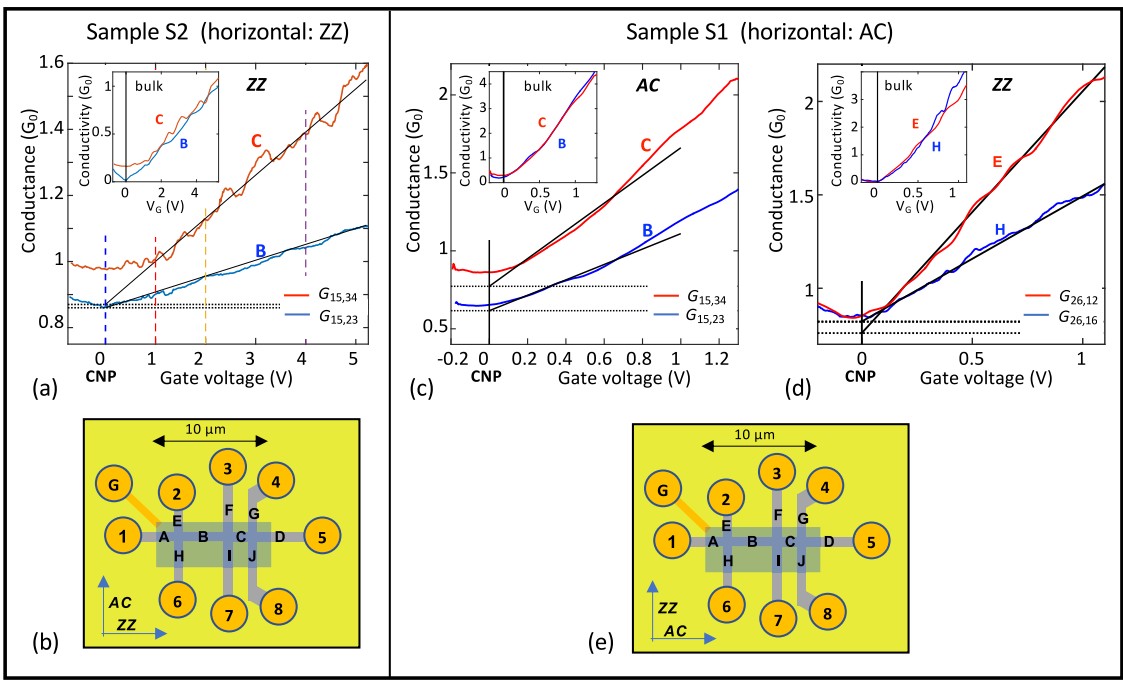

**Fig. 3 | Demonstration of the 1 $G_0$ edge state. a, b** Sample S2, axis along ZZ orientation, width = 1 μm. **a** 4-point conductances $G_B$, $G_C$ of segments B and C ($L_B$ = 4.5 μm, $L_C$ = 2.5 μm). The residual conductance $G^{res}$ (i.e. the edge state, dashed lines) is determined from extrapolation of $G(V_G)$ to $V_G$ = 0 (black lines) from which we find $\gamma_{ZZ}$ > 50 μm. See text for discussion of contact resistances and junction resistances. Inset shows that the bulk conductance $\sigma_{ZZ}$ for segments B and C are essentially identical, as expected. **b** Schematic diagram of S2: contacts (gold), segments (bold letters), and gate (blue rectangle). Ungated graphene ribbon sections have a charge

density $n_O \approx -2 \times 10^{11}$ cm$^{-2}$ ($\sigma_O \approx 1\ G_O$). **c–e** Sample S1, width = 740 nm, axis along AC direction. **c** 4-point conductances $G_B$, $G_C$ of segments B and C (AC segments, $L_B$ = 3.7 μm, $L_C$ = 1.7 μm), from which we find $\gamma_{AC}$ = 6 μm, $R_J$ = 0.08 $R_O$. **d** 2-point conductances $G_E$ and $G_H$ (vertical ZZ segments, $L_E$= 0.77 μm, $L_H$ = 1.9 μm). $G_E$ and $G_H$ converge at CNP which indicates that $\gamma_{ZZ}$ >> segment lengths $L_{E,H}$. Insets indicate that there is no significant anisotropy in $\sigma_{ZZ}$ and $\sigma_{AC}$. **e** Schematic diagram of S1. Ungated graphene ribbon sections have a charge density $n_O \approx -10^{12}$ cm$^{-2}$ ($\sigma_O \approx 1.5\ G_O$).

Figure 3a shows 4-point conductance measurements of ZZ ribbon segments **B** and **C** of S2: $G_B(V_G)$ (blue) and $G_C(V_G)$ (red). Scaling demands that $C_{CB} = G_C/G_B$ is strictly constant ($C_{CB} \approx L_B/L_C = 2$), however, $C_{CB}(V_G)$ varies from ≈1 to 1.5. Similarly, for AC ribbon segments **B** and **C** of S1 (Fig. 3c), $C_{CB}(V_G)$ varies from about 1.3 to 1.6. In both samples, the conductance reaches a minimum, but it does not vanish at the charge neutrality point (CNP). A disorder induced non-zero minimum conductivity $\sigma_{min}$ that obeys the scaling law is often seen in exfoliated graphene[14,72–74]. In those cases, extrapolating the measured conductance away from the rounding at CNP to $n$ = 0 gives $G$ = 0 (SI1). But here (Fig. 3a, c) it does not, and a large residual conductance $G^{Res}$ remains (dashed lines): for S1, $G_B^{Res}$ = 0.62 $G_O$, and $G_C^{Res}$ = 0.77 $G_0$; for S2, $G_B^{Res}$ = 0.83 $G_O$, and $G_C^{Res}$ = 0.85 $G_O$.

The residual conductance is immediately explained in the Landauer formulism (see, for example, ref. 53) where in general the conductance of a graphene ribbon with a ballistic edge state and a diffusive bulk (ignoring coherence effects), can be written as

$$G = G_{edge} + G_{bulk} = G_e(1+L/\lambda)^{-1} + \sigma W/L \qquad (1)$$

where $G_e$ is predicted to be 2 $G_0$ for a non-polarized graphene edge state and 1 $G_0$ for a polarized edge state[20,21,47,48,75–78]; the edge state mfp $\lambda$ is expected to depend on $E_F$.

Note that $GL/W$ (Eq. 1) depends on $L$. However, if we subtract $G_{B,C}^{Res}$ we find a segment-independent conductivity $\sigma_{B,C}=(G_{B,C}-G_{B,C}^{Res})L_{B,C}/W$. The fact that $\sigma_B$ and $\sigma_C$ coincide (Fig. 3a, c insets) shows that scaling is restored by this subtraction, which implies from Eq. 1 that $G_{B,C}^{Res}$ = $G_{edge}$ (see also SI4). Plugging the measured values in Eq. 1 gives $G_e$ = $G_{ZZ}$ = 0.86 $G_0$ and $\lambda_{zz}$ > 50 μm for the ZZ direction (sample S2), and $G_e$ = $G_{AC}$ = 0.95 $G_0$ and $\lambda_{AC}$ = 6 μm for the AC direction (sample S1). In contrast, the bulk mobility of S1, extracted from $\sigma$ is μ = 750 cm$^2$V$^{-1}$s$^{-1}$

which corresponds to a bulk mpf, $\lambda_{bulk}$ = 10 nm. This straightforward analysis of conventional 4-point measurements unambiguously shows that a 1 $G_0$ ballistic state exists at CNP.

The bulk conductivity $\sigma$ (Fig. 3a, c insets) is found by subtracting the residual conductance measured at $V_G$ = 0 (horizontal dashed lines), which implies that $\lambda$ in Eq. 1 does not depend on $V_G$ (if in fact it did, then $\sigma_B$ and $\sigma_C$ would not coincide). This independence contradicts the conventional band structure in Fig. 1a, where $V_G$ causes a rigid shift of the bands. For example, a $V_G$ = 1 V would shift $E_F$ by $\Delta E$ ≈ 110 meV so that $N_{OFl}$ would be $\Delta E$ below $E_F$ and would not participate in the transport which leaves only $N_{ODis}$. However, if $E_F$ were 110 meV then $N_{ODis}$ would mix with 4×$\Delta E$/ d$E$ = 220 bulk subbands (d$E$ ≈ 2 meV is the subband spacing for this 740 nm wide ribbon, Fig. 1a). Therefore $\lambda$ should decrease rapidly with increasing $V_G$, so that for large $V_G$, $G_{edge}$ should essentially vanish. In that case, we should have observed a pronounced ≈1 $G_0$ conductance peak at CNP, which is clearly not the case. Therefore Fig. 1a cannot represent the band structure for $V_G \neq 0$. This discrepancy is resolved below.

The measurements show that the edge state scatters at the junctions (see SI7)[79]. If the scattering were isotropic then the transmission probability through the junction would be $T_J$ =½ (see refs. 30, 53, 80) and a 4-point measurement would result in perfect 1 $G_0$ quantization. However at $B$ = 0, we find $T_J$ ≈ 0.45. The 10% discrepancy is discussed next. Figure 4a shows the conductance of Sample 2, ZZ, Seg. **B**, as a function of magnetic field $B$ for 4 values of $V_G$ at the vertical dashed lines in Fig. 3a. At CNP ($V_G$ = 0), the conductance increases to about 0.98 $G_0$ at $B$ = 3 T. Hence for $B$ > 3 T, $T_J$ ≈ ½ (see SI11 for additional measurements). Furthermore, as shown in Fig. 4b, the zero-field dip is strongly reduced with increasing temperature tending towards exact 1 $G_0$ quantization at CNP (see also SI11).

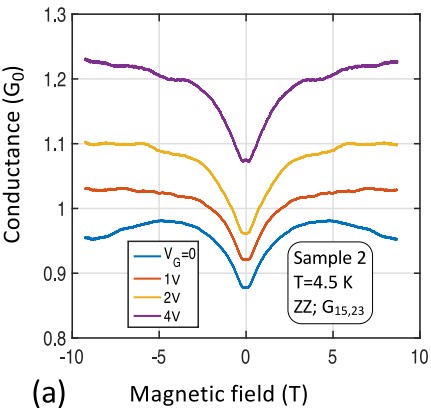

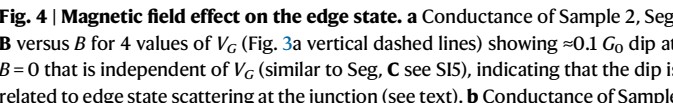

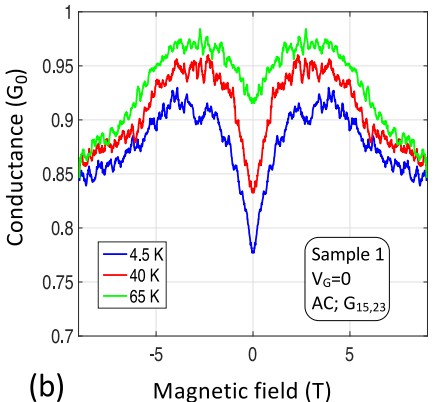

**Fig. 4 | Magnetic field effect on the edge state. a** Conductance of Sample 2, Seg. **B** versus $B$ for 4 values of $V_G$ (Fig. 3a vertical dashed lines) showing $\approx 0.1\ G_0$ dip at $B = 0$ that is independent of $V_G$ (similar to Seg. **C** see SI5), indicating that the dip is related to edge state scattering at the junction (see text). **b** Conductance of Sample 1, Seg. **B** at CNP for 3 temperatures, which shows that the conductance dip vanishes with increasing temperature. These properties suggest coherent scattering of the edge state at the junctions as explained in the text.

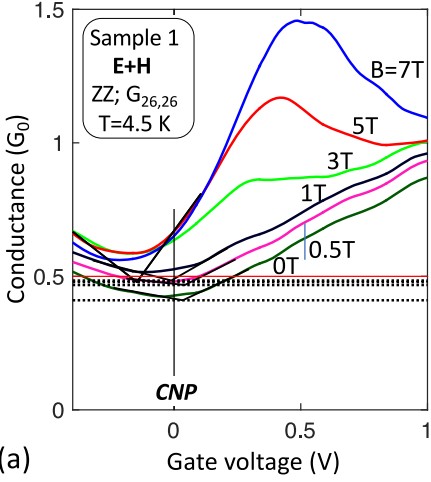

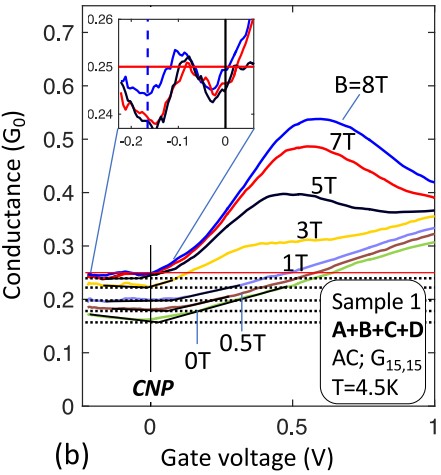

**Fig. 5 | Segmentation of the edge state.** The edge state conductance of a sequence of $N$ segments in series approaches $G_0/N$. **a** 2-point conductance $G_{EH}(V_G)$, ZZ, for various magnetic fields $B$. At CNP ($V_G = 0$), for large $B$, $G_{EH} \approx \frac{1}{2}\ G_0$, is consistent with a ballistic conductor with 1 isotropic scattering center (the junction). **b** 2-point conductance $G_{ABCD}(V_G)$, AC, for various $B$. At CNP ($V_G = 0$) the conductance at large $B$ is consistent with 4 conductors (each of conductance $\approx 1\ G_0$) in series, as expected for a $1\ G_0$ ballistic conductor with 3 isotropic scattering centers (3 junctions). In a magnetic field, the conductance increases to $\frac{1}{4}\ G_0$ (see inset for B = 5, 7, 8T) indicating essentially perfect quantization, which implies that $\lambda_{AC}$ diverges and $R_J$ vanishes.

The 3-point measurements of ZZ Segs. **E** and **H** of Sample 1 (Fig. 3d) are consistent with the 4-point measurements above, showing $G = 0.85 \pm 0.02\ G_0$ at CNP (taking into account the junction transmission $T_J$ and contact resistance, $R_c \approx 1.5$ kΩ). Figure 5a shows 2-point measurements of $G_{EH} = G_{26,26}$ as a function of $V_G$ (S1 along the ZZ direction) and Fig. 5b shows $G_{ABCD} = G_{15,15}$ (S1 along AC direction) for a range of magnetic fields. For $B = 0$, the residual conductance $G^{Res}_{ABCD} = 0.15\ G_0$ ($R^{Res}_{ABCD} = 6.6\ R_0$). If $T_J = \frac{1}{2}$ then $R^{Res}_{ABCD} = (4 + L_{ABCD}/\lambda_{AC})R_0$[30,53], so that $\lambda_{AC} = 6.3$ μm. For high magnetic fields the edge state conductances, $G_{EH}\ (V_G = 0)$ and $G_{ABCD}(V_G = 0)$, are very close to $\frac{1}{2}\ G_0$ and $\frac{1}{4}\ G_0$ respectively, as expected for a $1\ G_0$ ballistic conductor with respectively 1 and 3 isotropic scattering centers[30,53] (see also SI5). In addition, a magnetic field $B \geq 2$ T significantly increases $\lambda_{AC}$, i.e., it reduces backscattering at defects, reminiscently of the destruction of coherent backscattering by a magnetic field[53]. These observations confirm that the $1\ G_0$ ballistic edge state scatters at junctions, even in large magnetic fields, in contrast with quantum Hall edge states.

## 1D edge state

One might expect that edge state currents will follow the graphene periphery so that in general the conductance measured along the top edge of a segment should be unrelated to the bottom edge. However, the conductance measured along the top and bottom edges are identical, as shown for Seg. **B** of sample 2 (Fig. 6a), and Seg. **A** of sample 1 (Fig. 6b). Since only the edge state contributes to transport at CNP ($V_G = 0$) this demonstrates that edge state transport involves both sides of the ribbons equally, consistent with its theoretically expected 1D nature. Hence, the edge state forms a network of 1D single channel ballistic conductors with nodes at the junctions (see SI7).

## Vanishing edge state Hall voltage

The Hall resistance, $R_{Hall} = V_H/I_0$ (Fig. 7a) is determined from the Hall voltage $V_H = V_{26,15}$ and current $I_0$. Figure 7c plots $n^* = B/eR_{Hall}$ (see also SI6). For large $V_G$, $n^*$ is linear and independent of $B$, as expected in the diffusive limit where $n^* = n$. From this we derive the gate efficiency $C = 1.3 \times 10^{12}$ cm$^{-2}$V$^{-1}$ that agrees with the expected

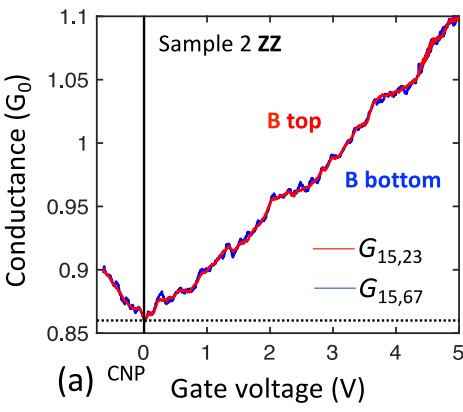

**Fig. 6 | Demonstration of 1D edge state transport. a** 4-point conductances of sample 2, Seg. **B** measured from the top ($G_{15,23}$; red) and the bottom ($G_{15,67}$; blue); **b** 3-point conductances of sample 1, Seg. **B** measured from the top ($G_{15,12}$; red) and the bottom ($G_{15,16}$; blue). The perfect overlap shown in the two cases is consistent with a 1D edge state that involves both edges coherently and not with a top edge that is independent of the bottom edge which would cause the conductances, especially at CNP to be very different (see text).

$C = \varepsilon_0\kappa/ed = 1.5 \times 10^{12}$ cm$^{-2}$V$^{-1}$, where $d = 30$ nm is the aluminum oxide gate dielectric thickness and $\kappa = 9$ is its dielectric constant. The thick black dashed line in Fig. 7a–c, corresponds to $(V_G - V_0^+) = n^*/C - \hbar v_F\sqrt{\pi n}/e$, where the second term represents the quantum capacitance correction[81]. The gate voltage $V_G = V_0^+ = 0.196$ V locates the conduction band edge (dashed blue line); it is significantly displaced from $V_G = 0$. Likewise, the valence band edge is approximately located at $V_G = -V_0$ which indicates an energy gap.

The edge state is ballistic in the segments ($\lambda \gg L$) and therefore it is effectively decoupled from the bulk. While edge state and the bulk thermalize at the contacts and back scattering occurs at the junctions, it is not clear whether inelastic edge state-to-bulk states scattering also occurs at the junctions. As shown below, Hall measurements indicate that this does not occur. This means that the edge state network is decoupled from the bulk network; the two only interact at the metal contacts. Consequently, an applied current $I_0$ is divided between the edge, $I_e$, and the bulk $I_b$:

$$I_0 = I_e + I_b;$$
$$I_e = I_0 G_{edge}(G_{edge} + G_{bulk})^{-1} \qquad (2)$$

Since edge state is a quantized non-degenerate ballistic graphene state, conventional theory predicts that it should generate a Hall voltage $V_{He} = \pm I_e R_0 \text{sgn}(B)$[48,82], where the sign depends on whether the edge state carrier is an electron or a hole. On the other hand, the diffusive bulk Hall voltage is $V_{Hb} = I_b B/ne$. As shown in SI8 the measured Hall voltage $R_{Hall}^m$ is easily calculated. Summarizing, $V_{He}$ and $V_{Hb}$ drive a circulating Hall current along the transverse arm, following the bulk, from the bottom to the top contact, and back, following the edge state. Consequently, for a cross with arms of equal length $L$ we expect that $R_{Hall}^m$ is given by Eq. 3a (see SI8 for details).

$$R_{Hall}^m = R_{Hb} + R_{He} = \frac{B}{ne}(R_b/R_e + 1)^{-2} + R_0(R_e/R_b + 1)^{-2} \text{ (a)}$$
$$R_{Hall}^m = \frac{B}{ne}(R_b/R_e + 1)^{-2} \text{ (for } R_{He} = 0\text{) (b)} \qquad (3)$$

where $R_e = R_0(1 + L/\lambda)$ and $R_b = L/(\sigma W) = L/(ne\mu W)$ are the edge state and bulk resistances of the arms. Equation 3b corresponds to an edge state with no Hall effect ($R_{He} = 0$).

Figure 7b plots $R_{Hall}/(R_0 B)$, the Hall resistance in units of $R_0$, versus $V_G$, which should be independent of $B$ in the diffusive limit. Indeed, for $B < 0.5$ T, we find that $R_{Hall}/(R_0 B)$ is independent of $B$ and accurately follows the diffusive limit for $V_G > 0.4$ V. However, it saturates at $\approx 0.2$ T$^{-1}$ for $V_G = 0.18$ V and reduces to 0 at CNP. Figure 7b inset shows that Eq. 3a does not agree with the data at all. On the other hand, Eq. 3b (Fig. 7b inset) reproduces all of the observed features. This shows that the edge state does not generate a Hall voltage (see also SI6). While a disordered electron-hole puddle state can produce a vanishing Hall voltage at CNP[47,53] and occasionally a quasi-quantized conductivity[14,47], it does not produce a ballistic state and certainly not one with a 1 $G_0$ conductance. Hence, the edge state is not the conventional graphene subband that was predicted[47,48].

Summarizing, we have shown that the edge state is an unconventional one dimensional 1 $G_0$ ballistic state that is insensitive to the gate voltage and does not generate a Hall voltage at any magnetic field.

## The anomalous quantum Hall plateau

Graphene ribbon theory[38] predicts that the $N = 0$ subband (Fig. 1a) in general (not only for zigzag ribbons) is characterized by a flatband over 1/3 of the Brillouin zone and a linearly dispersing band, $N_{0Dis}$. The flatband produces a large density of states at $E = 0$ at the ribbon edges, which has been experimentally observed in chemically produced ribbons[60], in sidewall ribbons[43], and again here in nonpolar epigraphene ribbons (Fig. 2f), however, it is not seen in patterned exfoliated graphene, which correspondingly does not have an edge state (SI1). The graphene edge state is predicted to develop from the flatband ($N_{0Fl}$), while a 2 $G_0$ quantum Hall state[48,82] corresponding to Landau level $LL_0$ is expected to evolve from $N_{0Dis}$.

Both the edge state and the bulk magnetoconductance are substantially insensitive to $B$ (note that $B\mu_{Bulk} < 1$), therefore only $N_{0Dis}(V_G, B)$ can have a significant $B$ dependence. Since $G^{0Dis}$ is diffusive (see above), it is small at $B = 0$ so that $G^{0Dis}(V_G, B) = G(V_G, B) - G(V_G, B = 0)$. Figure 8a plots $G^{0Dis}(V_G, B)$ vs $V_G$ for various $B$ for Seg. **H**. Note that $G^{0Dis}_H(V_G, B)$ appears to saturate near 1.6 $G_0$ (solid black line). Likewise, Fig. 8b plots $2G^{0Dis}_{EH}(V_G, B)$ for Seg. (**H+E**) (the factor 2 helps to compare the conductance of the two segments in series, $G_{H+E}$, with that of the single segment, $G_H$, see also Fig. 5a). It saturates near 1.8 $G_0$. Both cases are consistent with the 2 $G_0$ conductance of the zeroth Landau level $LL_0$ in the quantum Hall regime, thereby demonstrating that $N_{0Dis}$ causes the conductance bumps (see also SI10) The associated anomalous quantum Hall plateau (SI6) is explained below.

If we assume that the edge state also shunts the quantum Hall voltage, then the calculation leading to Eq. 3b (SI8) also predicts that the measured Hall resistance will be:

$$R_{Hall}^{Pred} = R^{0Dis}\left(R^{0Dis}/R_0 + 1\right)^{-2} \qquad (4)$$

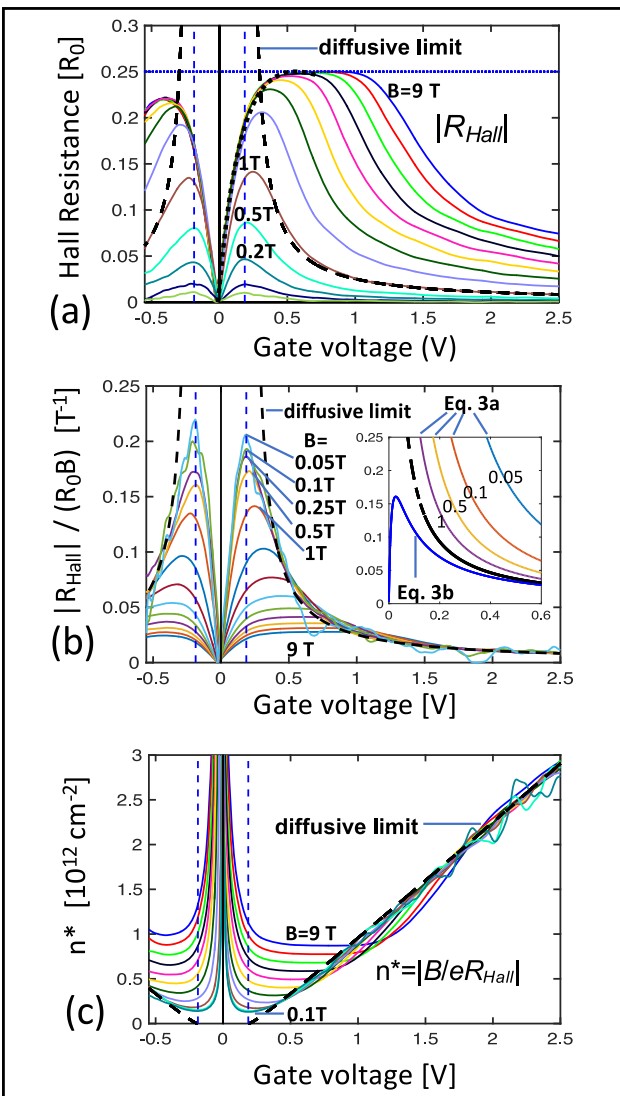

**Fig. 7 | Absence of an edge state Hall voltage. a** Hall resistance $R_{\mathrm{Hall}} = |R_{26,15}|$ (in units of $R_O$) versus $V_G$ for $0.02\,\mathrm{T} \le B \le 9\,\mathrm{T}$ shows an anomalous quantum Hall plateau at $\approx 0.25\,R_O$ for $B > 3\,\mathrm{T}$, consistent with a quantum Hall plateau from the dispersing branch $N = 0$ subband that is shorted by the edge state (see Eqs. 3b, 4). The black dashed lines is the diffusive bulk limit for $B = 1\,\mathrm{T}$. **b** The normalized Hall resistance $|R_{\mathrm{Hall}}/R_O B|$ converges to the diffusive bulk limit for $B < 0.5\,\mathrm{T}$ and $V_G > 0.4\,\mathrm{V}$. For $V_G < 0.4\,\mathrm{V}$ and $B < 0.1\,\mathrm{T}$ it becomes independent of $B$ and saturates at $\approx 0.2\,\mathrm{T}^{-1}$ and then decreases to 0 independent of $B$ as $V_G$ decreases to 0. This behavior is predicted in Eq. 3b (inset); Eq. 3a (inset) does not correspond at all, showing that the edge state does not generate a Hall voltage. **c** Effective charge density $n^* = |B/eR_{\mathrm{Hall}}|$. For small $B$ and/or large $V_G$, $n^*$ converges to the diffusive limit (thick black dashed line) which is corrected for the quantum capacitance. A significant gap $\Delta V_G = 0.17\,\mathrm{V}$ is observed which indicates a band gap which however cannot be quantified by this measurement.

Where $R^{\mathrm{ODis}} = 1/G^{\mathrm{ODis}}$. In fact, $R_{\mathrm{Hall}}^{\mathrm{Pred}}$, calculated from Eq. 4 using the experimental $G^{\mathrm{ODis}}{}_H(V_G,B)$ and $G^{\mathrm{ODis}}{}_{EH}(V_G,B)$ shown in Fig. 8c, d are remarkably similar and similar to the measured $R_{\mathrm{Hall}}$ (Fig. 7a), which verifies Eq. 4. Hence we conclude that the quantum Hall-like plateau is the graphene $R_O/2$ quantum plateau that is shunted by the edge state (see also SI6).

### Pinning of the edge state at $E = 0$

In neutral graphene ribbons the 0-DoS peak is half filled with $n_0 \approx 5 \times 10^8$ empty states per $\mathrm{cm}^2$ at $E = 0$ at the graphene edge[20]. Conventional condensed matter principles[52] dictate that the 0-DoS

peak will pin $E_F$ at $E_F = 0$. As long as the 0-DoS peak is not saturated, it will deplete gate-induced charges near the edge as theoretically verified[51] in simulations of 5–10 nm wide ribbons with a 2 nm high-κ dielectric top gate. to produce a Schottky barrier. The resulting electric fields bend the bands near the edge so that in the bulk of the ribbon the Dirac point is at $|E_D| = \hbar v_F \sqrt{\pi n}$ (Fig. 1b, d)[47,82]. The effective width of the depletion region is about half the dielectric thickness $d = 30\,\mathrm{nm}$ (ref. 51, Eq. 11). The 0-DoS peak saturates for $n_{\max} > 10^{13}\,\mathrm{cm}^{-2}$ (i.e., for $V_G > e/a_0 \varepsilon_O \kappa = 8\,\mathrm{V}$, with $a_0 = 2.5\,\text{Å}$ the graphene lattice constant) which is beyond our experimental range. Since we concluded that the annealed non-zigzag ribbons also terminate with acene atoms[67], they should have a similar $n_{\max}$.

This implies that the flatband pins the Fermi level that remains at E=0 along the entire graphene edge (excluding contacts), independent of $V_G$ for the gate voltages used here. These properties perfectly fit the characteristics of the edge state, which confirms that the edge state is pinned at $E=0$ (i.e., independent of $V_G$) and protected by a Schottky barrier.

An excited 1 $G_O$ edge state $\Delta E \approx 10\,\mathrm{meV}$ above the ground state is also observed (SI10) which is probably related to the recently observed $\approx 0.1\,\mathrm{eV}$ gap in narrow sidewall ribbons[43] so that, as expected, this energy gap is tuned by the ribbon width[30,36,37,43,45].

To qualitatively illustrate the band structure (Fig. 9), we use a staggered potential to produce a gap[45], onsite potentials near the edge for the electrostatic edge potential, and couple the upper and lower $N = 0$ states as for a ferromagnetic edge state[83]. For small $n$, the bulk Hall conductance (Fig.7c dashed line) vanishes at the conduction and valence band edges (Fig. 9b, d) however, since it is shunted by the edge state, $R_{\mathrm{Hall}}$ does not diverge. For large $n$ the bulk conductance dominates, and the Hall resistance converges to the diffusive limit (Fig. 7b dashed line). $N_{\mathrm{ODis}}$ becomes the $LL_0$ quantum Hall-like state in a magnetic field[82] (Figs. 7a, 8). Mixing with the bulk states (Fig. 9a, e) ultimately disrupts the quantum Hall-like state as $|V_G|$ increases.

## Discussion

Several observed edge state properties seen here for the first time are consistent with theoretical predictions. However, the observation of a non-degenerate edge state that does not generate a Hall voltage is not predicted. Single-sided edge state transport, as recently proposed for sidewall ribbons[43,84] is ruled out since both sides of the ribbons participate in the transport. Consequently, since the edge state comes from the flatband at $E = 0$ that is composed of electrons and holes, it is natural to consider a quasiparticle that is half electron and half hole, where both components follow the same path (even in a magnetic field) but in opposite directions. Interestingly, this quasiparticle is effectively charge neutral so that it does not generate a Hall voltage, but it will carry a current, half carried by the electron and half by the hole. If in addition this edge state is spin polarized as it is in sidewall ribbons, (see SI13), then the proposed quasiparticle will be a spin ½ fermion with no Hall voltage and with a conductance $G = 1\,G_O$, as observed. Note that the unusual quasiparticle described here is technically a spin polarized non-chiral fermion of which the Majorana fermion is an important example. In fact the edge state could be a Majorana fermion or a related new quasiparticle.

The excellent transport properties of the epigraphene edge state properties suggest a wide variety of device architectures, besides ultra-high frequency epigraphene field effect transistors[85]. For example, a graphene nanoribbon supplied with top and side gates is predicted to have impressive switching characteristics[22] (SI14). The spin polarized edge state (SI13)[86] provides an important step toward spintronics applications[87,88]. Epigraphene ribbon constrictions have non-linear properties that may be modulated with a gate[89,90]. The excited edge state observed here and in sidewall ribbons may be used in graphene ribbon tunneling transistors[91]. In fact, the silicon carbide substrate itself, which is compatible with THz electronics and in which qubits

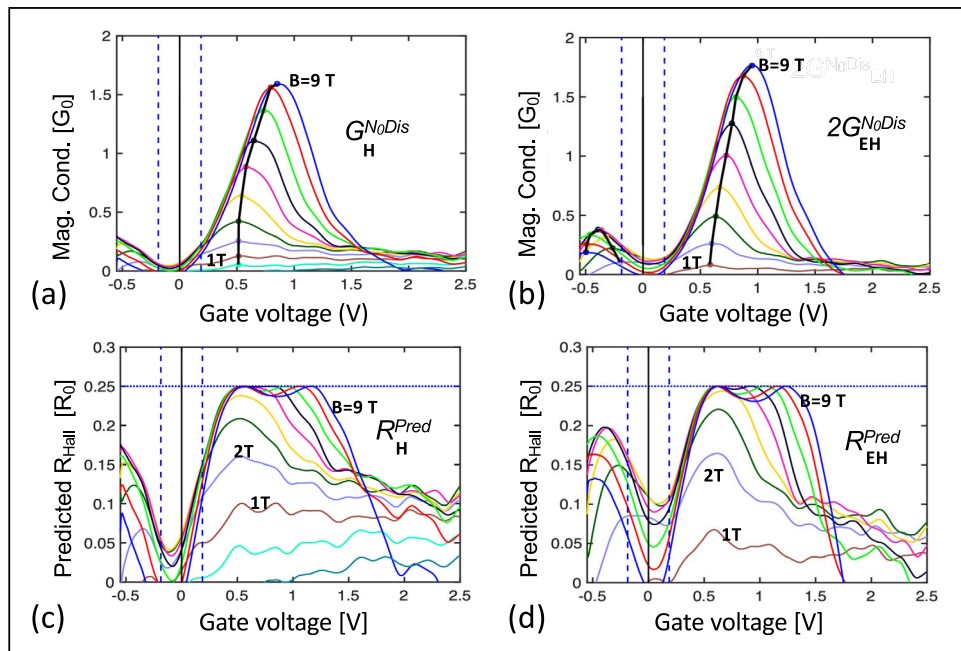

**Fig. 8 | Anomalous quantum Hall plateau. a** Magnetoconductance of $N_{0Dis}$, in units of $G_0$ for Seg. **H**. **b** Twice the magnetoconductance of $N_{0Dis}$ for Seg. (**E+H**), see text. Note the saturation near $2G_0$ as expected for $N_0$ in the quantum Hall regime. **c, d** Calculated quantum Hall resistance of Seg **H** and Seg. (**E+H**) from Eq. 4, that predicts the anomalous $0.25\,R_0$ plateau from the magnetoconductance, see text. Note the remarkable resemblance between **a** and **c**, and **b** and **d**, which further strongly supports Eq. 4 and hence the vanishing of the edge state Hall voltage.

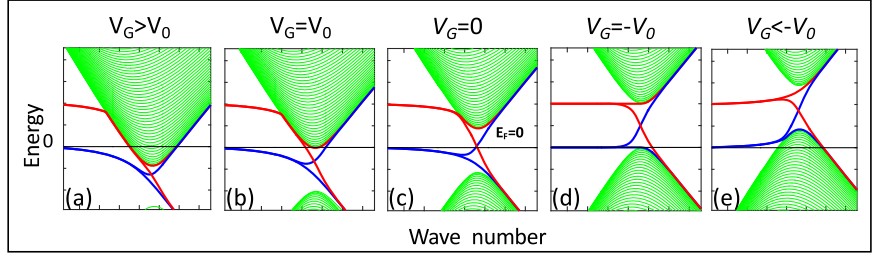

**Fig. 9 | Tight-binding model for gapped graphene with an edge state pinned at E=0 for various gate voltages, $V_G$. a–e** Conduction and valence band (green) and split edge state (blue and red) for increasing $V_G$. For $V_G = V_0$ in (**b**), the Fermi level grazes the bottom of the conduction band, which is where the Hall voltage of the bulk conduction band vanishes. For $V_G = -V_0$ in (**d**) $E_F$ grazes the top of the conduction band where the bulk valence band Hall voltage vanishes. The Hall voltage of the $N = 0$ subband crosses 0 midway between these two in (**c**).

may be realized[92], may be incorporated in switching schemes[93]. Moreover, like carbon nanotubes[9] and graphene quantum Hall states[94,95], the zero-mode nanostructures will be phase coherent at low temperatures[13], enabling graphene quantum interference devices[9,95] so that ultimately quantum coherent structures may be realized that could be used for quantum computing[94]. Moreover, silicon-on-epigraphene methods[33] have already been demonstrated which allows integration with silicon electronics.

While more experimental and theoretical work needs to be done, the constellation of properties demonstrated here, on a nano-patternable substrate, shows that nonpolar epigraphene is an ideal platform for essentially dissipationless integrated zero-mode graphene nanoelectronics as originally envisionned[6,12].

## Methods
### Sample fabrication
Non-polar wafers were produced in-house from commercial bulk single crystal 4H-SiC rod, by cutting them along directions corresponding to the sidewall facets ($\bar{1}10n$), $n = 5$. The wafers were then CMP polished. Graphene samples were prepared using the Confinement Controlled Sublimation method[25] in a graphite crucible provided with a 0.5 mm

hole, under various growth conditions depending on the crucible condition, all producing monolayer epigraphene: Sample S1 (Figs. 3–8) in a 1 atm Ar atmosphere at 1550 °C for 30 min followed by 1650 °C for 2 h; S2 in 1 atm Ar atmosphere at 1500 °C for 30 min followed by 1600 °C for 1 h; the sample in Fig. 2b, c, e was grown in vacuum in a face-to-face configuration at 1550 °C for 20 min, and the sample of Fig. 2d and SI2 in a 1 atm Ar atmosphere at 1550 °C for 30 min followed by 1650 °C for 15 min.

### Sample patterning
Sample S1 (Figs. 3–6) was patterned using conventional lithography methods. An alumina protecting layer was first deposited on graphene immediately after annealing in vacuum at 1000 °C for 30 min, where Al was first evaporated (0.5 Å/s) in $\approx 5 \times 10^{-5}$ mb oxygen atmosphere. Hall bars were e-beam patterned with a bilayer MMA/PMMA resist and provided with an alumina coating. The graphene/alumina was dry-etched in $BCl_3$ plasma (ICP) through a PMMA/PPM e-beam patterned mask. Buffered HF was used to provide openings in the alumina for contacts. E-beam evaporated Pd/Au was used for contacts and the gate electrode (see Fig. S4). For sample S2 a Hall bar patterned $Al_2O_3$ coating serves as a mask in a RIE $O_2$ plasma etch followed by stripping

$Al_2O_3$ in an Al etchant and a thermal anneal at 1200 °C for 15 min. Following patterning provides Au/Pd side contacts and top gate (15 nm of $Al_2O_3$ as dielectric), see Fig. S5.

## Experimental measurements

Transport measurements were performed in a 1.6–420 K cryocooler, provided with a 9 Tesla magnet. Voltages were sequentially measured by eight lock-ins (frequency < 21 Hz), with low current excitation (from 1 to 10 nA). Cryogenic STM images were made in high resolution, AFM/STM[96] at the Néel institute, and at the TICNN using a RHK PanScan Freedom STM. Raman spectra were acquired with a high-resolution confocal Raman microscope system at an excitation wavelength of 532 nm. Room temperature ARPES measurements were performed at the CASSIOPEE beam line of the Soleil synchrotron, equipped with a Scienta R4000 analyzer and a modified Peterson PGM monochromator with a resolution $E/\Delta E = 70,000$ at 100 eV and 25,000 for lower energies. The 6 axis cryogenic manipulator is motorized. The ARPES sample was prepared ex-situ and cleaned under ultra-high vacuum conditions by flash heating it at 700 °C. The infrared (IR) magneto-spectroscopy measurements were carried out in reflection mode using a standard Fourier-transform IR spectroscopy technique (Bruker VERTEX 80 v) at liquid helium temperature. The IR light from a Globar source was delivered to the non-polar epigraphene through an evacuated light pipe, and the reflected light was guided to a Si bolometer away from the magnetic field center. All measurements were performed in Faraday geometry with the field applied perpendicular to the graphene.

## Data availability

Data are fully available upon request to the corresponding authors.

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

## Acknowledgements

Financial support was provided by the U. S. National Science Foundation-Division of Electrical, Communications and Cyber Systems (No. 1506006) and NSF-Division of Material Research No 1308835. C.B and V.P. acknowledge funding from the European Union grant agreements No. 696656 and No. 785219. This work was also made possible by the French American Cultural Exchange council through a Partner University Fund project and a Thomas Jefferson grant and by the Agence Nationale de la Recherche (No. ANR-19-CE24-0025). Financial support is acknowledged from the National Natural Science Foundation of China (No. 11774255), the Key Project of Natural Science Foundation of Tianjin City (No. 17JCZDJC30100), and the Double First-Class Initiative of Tianjin University from the Department of Education in China. W.d.H. and L.M. thank Prof. Jiajun Li for his unwavering support in creating the TICNN center. The magneto-infrared spectroscopy measurement was supported by the U.S. Department of Energy (grant No. DE-FG02-07ER46451) and performed at the National High Magnetic Field Laboratory, which is supported by NSF Cooperative Agreement No. DMR-1644779 and the State of Florida. We thank Evangelos Papalazarou as well as François Bertrand and Patrick Lefebvre for their help with the ARPES measurement at the synchrotron Soleil-Cassiopée beam line. Nikolay Cherkashin is thanked for polishing preliminary nonpolar SiC chips, and John Hankinson and Owen Vail for spin transport measurements, Noel Dudeck for Fig. S21b, c. W.d.H. thanks M. Crommie, C. Delerue, P.N First, L. Levitov, M. Pustilnik, E. Rossi, and O. Yazyev for helpful discussions.

## Author contributions

V.P., C.B., Yiran H., Yue H., and G.N. performed transport measurements and device patterning (Figs. 3–8, S2–14, S20), graphene growth (Figs. 2–8, S2–S20) and sample characterization (Figs. 2a, b, S15c). L.M., K.Z., P.J., J.Z., and C.S., fabricated and characterized nonpolar face SiC wafers, performed graphene growth (Fig. 2a-inset, 2f) and high resolution STM and STS measurements (Fig. 2a-inset, 2f); Z.J., Y.J., and T.Z. performed the IR spectroscopy (Fig. 2d). C.B., V.P., and A.T. performed the ARPES experiments (Figs. 2c, S15a-b); D.W., A.d.C., and C.W. performed STM and STS experiments (Fig. 2e and Fig. SI15e-f). KW provided theoretical support and Figs. 1b, S17 and W.d.H. Figs. 1a, 1c, 1d, 9. C.B. and W.d.H. directed the Atlanta based experiments. L.M. and W.d.H. directed the TICNN efforts. W.d.H. is primarily responsible for the analysis and interpretation.

## Competing interests

The authors declare no competing interests.
