## [Peer Review File · Nature Communications]

An epitaxial graphene platform for zero-energy edge state
nanoelectronicsREVIEWER COMMENTS

Reviewer #1 (Remarks to the Author):

The manuscript reports ballistic transport via the zero-mode state of zigzag edges in the so called epi-graphene, which is epitaxially grown on non-polar faces of silicon carbide substrate through thermal annealing. The authors claim that the chirality of graphene ribbon and the edge states are extremely robust to general lithographic processing, making epi-graphene a viable platform for graphene nano-electronics. The authors also reports peculiar transport properties, suggesting that the edge state transport involves unconventional quasiparticle. The results are highly interesting, worthy to share with the community. I would suggest a major revision before accepting the manuscript for publication. My comments and suggestions are detailed below:

1. I understand that processing of monolayer graphene on SiC with well-defined edges are already established by previous work of the authors. However, for the completeness of the manuscript and for the convenience of the readers, the authors shall give information about the growth of epi-graphene with specific edges;
2. It is unclear for me what do the authors mean by saying “novel properties point to an unconventional charge carrier that is half-electron and half-hole”. Are the authors talking about “spin” and not “charge”? A half-electron and half-hole would make the quasiparticle charge neutral, then where come the current and voltage?
3. Fig.1 shows that a Schottky barrier at the edge isolates the edge state from that of bulk channel (film or ribbon). What is the width of the Schottky barrier? For a wide graphene ribbon, like that in Fig.1 (740nm), the Schottky barriers can isolate very well the edge states from that of the bulk. However, for ultra-narrow ribbons which are more interesting for nano-electronics, if the width of the barrier is not short enough, then the isolation between the edge states and that of the bulk will be weakened. Do the authors have any comment or explanation on this?
4. The last paragraph in page 3 is a bit confusion for me. The authors first state that “only thermally annealed epi-graphene has an edge state”, followed by method 1, edge is robust against RIE, because etching is followed by an annealing process at ~1200C. The authors continued by saying that the edge states are robust against ICP and e-beam patterning, but such processes are not followed by annealing. Normally such processes may cause roughening of the edges, making the chirality of the edges mixed. It seems that the authors believe that, besides etching effects, such processing involving high energy ion collisions can also serve as fast high temperature pulse annealing, so that during cooling, zigzag edges are restored as a result of interaction of carbon atoms with the substrate. Are there experimental evidences on that? I would suggest that the authors clarify this and rephrase this part to make it clearer.
5. In Fig.3, demonstration of the 1 G0 edge states along the zigzag edges which is the longer side of the rectangular shaped hall-bar. Is the shorter side of the sample with armchair edge or with mixed chirality? If they are arm-chaired, how are they made experimentally?
6. In Fig.3, some measurements were carried out in two or three terminal configuration. As we know that the conductance of two-terminal graphene devices in the QH regime strongly depends on the geometry (Phys. Rev. B 80, 045408 (2009)), how could the authors extract the contribution of bulk conductance from the overall conductance? It is very important to identify the contribution of 1G0 edge state. Please clarify.
7. For graphene on non-polar (1-10n), n=5 surface of SiC, the graphene is charge neutral. Does this mean that the graphene is decoupled from the substrate, except for carbon atoms at the edges? I would suggest that the authors illustrate the terminal Si and C atoms on the (1-10n), n=5 surface, and give surface roughness information of graphene on such non-polar surface of SiC.

Reviewer #2 (Remarks to the Author):

This paper is about ballistic transport via edge modes in graphene nanoribbons. This group has previously reported ballistic edge mode transport in graphene nanoribbons that grow in trenches on SiC so that the nanoribbon edges terminate at trench sidewalls. This growth/trench scheme is problematically not easily amenable to complex circuit fabrication. The novelty here is that this same effect has now seemingly been obtained at the edges of lithographically patterned nanoribbons on SiC. If true, this effect could enable such ballistic transport to be more rationally exploited in devices, enable new types of devices that leverage novel physics, and have significant impact, as the paper elaborates on extensively.

1) The “nano”ribbons in the main two samples characterized have widths of about 1 micrometer. The paper would be much more convincing and therefore have a much higher impact if this effect could be demonstrated on a much wider range of nanoribbon widths and lengths. Instead of the cumbersome discussion of the ratios of the conductance of segments B and C and how this ratio only make sense once edge modes and bulk conductivity are separately considered, data is needed on a much more sizeable range of W and L , especially very narrow W where the bulk conductivity should drop precipitously without changing edges’ ability to support edge modes.

2) Missing control data. The paper states “We find that only thermally annealed epigraphene structures have an edge state.⁵⁹” The paper needs to include control data of nanoribbons with same dimensions, comparing with and without the thermal annealing, that is believed to repair the edge-mode transport.

3) The paper makes overly broad claims about literature and field. For example, the paper states, “This makes epigraphene the only technologically viable graphene nanoelectronics platform that has the potential to succeed silicon nanoelectronics.” Firstly, the paper seems to define “graphene nanoelectronics platform” as something very specific related to ballistic transport and edge modes whereas most others in the community likely would have a different picture in mind for the words “graphene nanoelectronics platform” such as any use of semiconducting graphene nanoribbons or arrays of graphene nanoribbons or possibly even semiconducting carbon nanotubes integrated into field effect transistors that have a circuit-level energy-delay product less than Si. This is already possible with carbon nanotubes (see L.-M. Peng @ Peking Univ. or S. Mitra atd Stanford). Secondly, “In contrast lithographically patterned exfoliated graphene invariably has highly disordered graphene edges²⁷⁻²⁹ which effectively makes it unsuited for nanoelectronics.” There are multiple reports of edge-repair approaches (see early work from H. Dai, Stanford). The direct growth of surface-synthesized GNRs on dielectrics is now possible, which to many seems to open viable approach to semiconducting graphene and electronics that is not on SiC.

Other:

4) “The protected edge state has a mean free path that is greater than 50 microns” at what temperature? Please specify temperature of data presented in each figure.

Responses to Referees

Referee #1

“The manuscript reports ballistic transport via the zero-mode state of zigzag edges in the so called epi-graphene, which is epitaxially grown on non-polar faces of silicon carbide substrate through thermal annealing. The authors claim that the chirality of graphene ribbon and the edge states are extremely robust to general lithographic processing, making epi-graphene a viable platform for graphene nano-electronics. The authors also reports peculiar transport properties, suggesting that the edge state transport involves unconventional quasiparticle. The results are highly interesting, worthy to share with the community. I would suggest a major revision before accepting the manuscript for publication.”

We thank Referee #1 for his/her encouraging summary

1. “I understand that processing of monolayer graphene on SiC with well-defined edges are already established by previous work of the authors. However, for the completeness of the manuscript and for the convenience of the readers, the authors shall give information about the growth of epi-graphene with specific edges;”

The epitaxial alignment of the graphene with the single crystal SiC substrate is known and confirmed both with LEED and ARPES (Figs 2c, S15a-c). Therefore a cut with respect to the known crystallographic axes of the SiC substrate determines the chirality of the graphene edge. By carefully aligning the SiC chip in the lithographer we can accurately cut the graphene along a zigzag axis. A cut perpendicular to that will be along a graphene armchair axis. We included the following in the text:

“This is essential to accurately direct the lithography, however as discussed below, in general, even an annealed edge is generally not straight so that the actual edges do not have a well defined chirality.”

and

“Here we use the ZZ and AC designation to distinguish these two directions and not to indicate the actual graphene edge morphology, which for annealed edges is determined by their stability that involves the bonding to the substrate, mostly likely favoring acene bonded edges (See SI17).”

2. “It is unclear for me what do the authors mean by saying “novel properties point to an unconventional charge carrier that is half-electron and half-hole”. Are the authors talking about “spin” and not “charge”? A half-electron and half-hole would make the quasiparticle charge neutral, then where come the current and voltage?”

This is a critically important question which is partly answered in the first paragraph of the conclusions. Thank you for the opportunity to amplify this point. In fact our experiments are consistent with the current being carried by an electron going in one direction that equals the current of a hole propagating in the opposite direction. These separate currents have compensating Hall effects giving a net 0 Hall voltage. If the

currents were due to independent spin polarized ballistic single channel currents then the total conductance would be $2 e^2/h$. However we observe $1 e^2/h$, which is half this value. This factor of 2 implies that in fact these two currents are not independent but correlated to form a single spin polarized fermion that can be seen as half an electron and half a hole moving in opposite directions. This hypothetical quasiparticle is a non-chiral fermion. An example of exactly such a fermion is the Majorana fermion propagating from the source to the drain electrode, which is in fact charge neutral yet can carry a current with a conductance of $1 e^2/h$ and a spin $\frac{1}{2}$. We have added this description in the text. In addition we added:

“Note that the unusual quasiparticle described here is technically a spin polarized non-chiral fermion of which the Majorana fermion is an important example. The possibility that the edge state is in fact a Majorana fermion is not ruled out.”

This is an accurate statement as confirmed by theorists, however the claim that the edge state quasiparticle is in fact a Majorana cannot be made since theorists have advised us that there may be other possibilities.

3. “Fig.1 shows that a Schottky barrier at the edge isolates the edge state from that of bulk channel (film or ribbon). What is the width of the Schottky barrier? For a wide graphene ribbon, like that in Fig.1 (740nm), the Schottky barriers can isolate very well the edge states from that of the bulk. However, for ultra-narrow ribbons which are more interesting for nano-electronics, if the width of the barrier is not short enough, then the isolation between the edge states and that of the bulk will be weakened. Do the authors have any comment or explanation on this?”

This too is an important question. In fact the theoretical models by Guo et al, Ref.49 that first predicted the Schottky barrier, involved simulations of 5-10 nm ribbons supplied with a 2 nm high K dielectric, which is precisely in the range that is relevant for nanoelectronics. It is important to note that in these very narrow ribbons, the subband spacing of the bulk states is in the range of 0.1 eV compared to about 1 meV in 1 μm wide ribbons. This band separation significantly increases the isolation of the edge state in the narrow ribbons. In the text we added corresponding statement.

4. “The last paragraph in page 3 is a bit confusion for me. The authors first state that “only thermally annealed epi-graphene has an edge state”, followed by method 1, edge is robust against RIE, because etching is followed by an annealing process at $\sim 1200\text{C}$. The authors continued by saying that the edge states are robust against ICP and e-beam patterning, but such processes are not followed by annealing. Normally such processes may cause roughening of the edges, making the chirality of the edges mixed. It seems that the authors believe that, besides etching effects, such processing involving high energy ion collisions can also serve as fast high temperature pulse annealing, so that during cooling, zigzag edges are restored as a result of interaction of carbon atoms with the substrate. Are there experimental evidences on that? I would suggest that the authors clarify this and rephrase this part to make it clearer.”

Thank you for this question. The annealing effect alluded to by the referee has been theoretically investigated in a thorough paper titled “Tight-binding quantum chemical

molecular dynamics simulations for the elucidation of chemical reaction dynamics in SiC etching with SF₆/O₂ plasma,” Ref. 61. This indeed predicts very high local temperatures caused by the ion impacts similar to those that occur in conventional plasma welding but on the nanoscale. These high temperatures anneal the graphene edges by bonding them to the SiC as is the case for thermally annealed RIE etched edges. Note that the edges of graphene patches on SiC and edges of graphene on SiC substrate steps have been experimentally investigated using STM methods (T.T.N. Nguyen *et al.* Topological Surface State in Epitaxial Zigzag Graphene Nanoribbons. *Nano Lett* **21**, 2876-(2021), Figs 2b &3a) and it is found that they are in fact bonded to the substrate with acene (i.e. zigzag) bonds as we mentioned in the text. We have not yet been successful in performing similar very challenging atomic scale STM measurements of ICP etched graphene edges. As for the effect of rough edges on edge state transport, several authors have predicted, that edge state ballistic transport is quite insensitive to edge roughness and the resulting chirality mixing, as long as the edge atoms are properly terminated, see for example Refs 18, 36, 37. We have experimentally confirmed this in the paper.

5. “In Fig.3, demonstration of the 1 G₀ edge states along the zigzag edges which is the longer side of the rectangular shaped hall-bar. Is the shorter side of the sample with armchair edge or with mixed chirality? If they are arm-chaired, how are they made experimentally?”

The shorter side corresponds the armchair direction, however the direction of the cut does not necessarily correspond to the morphology of the edge. Specifically what we call an armchair edge is an edge that is cut along the nominally armchair direction as explained in our response to comment 1 above. However after lithography and annealing we find that the edges are not perfectly straight so that they are of mixed chirality, as mentioned above in response to (2). We find that the mean free path of a nominally armchair edge is much shorter than that of a nominally zigzag edge which indicates that the mixed chirality of the edge has a more significant effect in the armchair direction than in the zigzag direction. For reasons explained above, we have not been able to make the required atomic resolution images of these edges. But as a guide for discussion we have included a schematic diagram with explanation in the Supplementary material (Fig. S21).

6. “In Fig.3, some measurements were carried out in two or three terminal configuration. As we know that the conductance of two-terminal graphene devices in the QH regime strongly depends on the geometry (Phys. Rev. B 80, 045408 (2009)), how could the authors extract the contribution of bulk conductance from the overall conductance? It is very important to identify the contribution of 1G₀ edge state. Please clarify.”

We realize that our discussion of the transport measurements is quite dense. However the questions raised by the referee are treated there. From the phrasing of the question I assume that the measurements at zero magnetic field are clear to the referee, however that measurements at high fields are not. The bulk has a small mobility $\mu \approx 700 \text{ cm}^2\text{V}^{-1}\text{s}^{-1}$ which is too small to support a quantum Hall state ($\mu B < 1$ at all measured B). Hence conventional quantum Hall effects play no role as far as the bulk is concerned which is in the diffusive regime for all fields and temperatures used here. Consequently

in our experiments, the magnetic field generates a conventional Hall voltage and the longitudinal conductance is slightly affected, as usual for a diffusive conductor.

In the QHE regime (applicable for high mobility graphene, not our case) the quantum Hall edge state is simply a single channel ballistic conducting channel from the source to the drain, however it has the unusual feature that this channel is not interrupted by the non-current carrying contacts. Consequently, a two-point measurement has a quantized resistance that is nominally the same as the Hall resistance, if the contact resistance is ignored. Moreover, there is no voltage drop between non-current carrying contacts on either side of the Hall bar.

In contrast at the charge neutrality point, with or without a magnetic field we observe the edge state forms a network of $1 G_0$ ballistic conductors from junction to junction (where the bulk has a small positive contribution). Hence we do not observe a quantum Hall state.

Away from charge neutrality we do see the bulk contribution which causes the linear rise in the conduction. As is conventionally done, we extrapolate the linear rise back to the charge neutrality point, which allows us to accurately determine the small bulk contribution to the conductance at the charge neutrality point mentioned above. Comparing 2, 3 and 4 point measurements made for all contacts available allows us to determine and account for the actual contact resistances using Ohms law as we describe. These contact resistances are found to be small and consistent with previous measurements.

We also find the resistance contribution from the junctions that vanishes in high magnetic fields and with increasing temperatures.

In short, measurements performed at the charge neutrality point have a small contribution from the bulk and from contacts that we can experimentally determine. In a magnetic field, at charge neutrality, we find a reduction of the junction resistance that is important only at low temperatures.

We do see a quantum Hall like state, which is caused by the dispersing branch of the edge state, not from the bulk, as explained in the text.

We thank the referee for pointing out the 2009 PRB paper by Williams et al. but we do not see effects that are described there. Moreover, Lev Levitov, who is the theoretical coauthor on that paper has studied our paper carefully, and made no mention of this paper to explain our results.

We made a few modifications to the text to make these points clearer there.

7. "For graphene on non-polar (1-10n), n=5 surface of SiC, the graphene is charge neutral. Does this mean that the graphene is decoupled from the substrate, except for carbon atoms at the edges? I would suggest that the authors illustrate the terminal Si and C atoms on the (1-10n), n=5 surface, and give surface roughness information of graphene on such non-polar surface of SiC."

The charge appearing on the graphene on the polar faces is induced by the charge density due to the dangling bonds of the C and Si surface atoms for the polar faces and not by chemical coupling to the substrate, according to Seyller et al (Ref 68). This substrate surface charge density is neutralized by passivating the surface with hydrogen for example. In our case we expect that there are fewer Si and C dangling bonds, if any, which apparently reduces the induced charge on the graphene. A similar effect is observed on the cubic 3C-SiC [S. Mammadov *et al.* Polarization doping of graphene on silicon carbide. *2d Mater* **1**, 035003 (2014).]

As for the roughness, the non-polar graphene surfaces are well ordered, as is evident from the LEED measurements (Fig. S15 b) and AFM images (Fig. S15f). The unreconstructed SiC (1-105) model surface the graphene is lying upon is shown in (Fig. S15 d). We believe that this information adequately addresses the Referee's question. We are reluctant to go beyond this basic structural description, because a more elaborate picture may give the impression that we have studied the actual reconstructed surface and the graphene/SiC interface which is well beyond our capabilities at the moment and the scope of this paper.

Referee #2

“This paper is about ballistic transport via edge modes in graphene nanoribbons. This group has previously reported ballistic edge mode transport in graphene nanoribbons that grow in trenches on SiC so that the nanoribbon edges terminate at trench sidewalls. This growth/trench scheme is problematically not easily amenable to complex circuit fabrication. The novelty here is that this same effect has now seemingly been obtained at the edges of lithographically patterned nanoribbons on SiC. If true, this effect could enable such ballistic transport to be more rationally exploited in devices, enable new types of devices that leverage novel physics, and have significant impact, as the paper elaborates on extensively.”

We thank the Referee for his/her kind summary.

(1) “The “nano”ribbons in the main two samples characterized have widths of about 1 micrometer. The paper would be much more convincing and therefore have a much higher impact if this effect could be demonstrated on a much wider range of nanoribbon widths and lengths. Instead of the cumbersome discussion of the ratios of the conductance of segments B and C and how this ratio only make sense once edge modes and bulk conductivity are separately considered, data is needed on a much more sizeable range of W and L , especially very narrow W where the bulk conductivity should drop precipitously without changing edges’ ability to support edge modes.”

We first of all appreciate that the Referee is not disputing our claim of a ballistic long mean free path edge state, but would like to see it demonstrated with more ribbon geometries. We thank the Referee for his/her suggestion to strengthen the paper, and we agree to a certain extent. We have added data on 50 measurements in 2 figures in the Supplementary Material, Section SI 16; however there is more to it, and we give a more comprehensive response here to justify our presentation in the text.

First of all, it is clear from the work here that we are observing the same edge state as was previously observed in the sidewall ribbons [Ref 27] whose nominal width is 40 nm and lengths were measured from 100 nm to 30 μm . We also have made 4-point measurements of 50 RIE etched Si face ribbons of 80-100 nm widths and of lengths ranging from 1-6 μm before and after thermal annealing (See SI16). The annealed ribbons have an edge state that is consistent with $1 G_0$ measured for a range of temperatures [Yue Hu, *The edge states of epitaxial graphene on SiC*, PhD dissertation thesis, Georgia Tech (2021)]. In that case the bulk charge density is about $5 \times 10^{12} \text{ cm}^{-2}$. So we hope that these measurements satisfy the criteria of the Referee.

The difficulty to determine the transport properties of the edge state and to accurately determine its conductance value is that it requires careful accounting of all of the spurious resistances related to the bulk, as well as finite mean free paths and temperatures effects that cause a significant spread in the raw measured conductances of nominally similar ribbons. In absence of a careful analysis, as presented in the main text, the claim of a $1 G_0$ ballistic edge state is significantly weakened. In fact, our earlier work failed to convince referees of this theoretically not predicted effect, precisely because of this. That is why we performed the more careful measurements of the two Hall bars and the thorough analysis to sort out the details so that we can provide the unambiguous results we show here.

In contrast to what the referee implies, our analysis involved all segments, not only Seg. B and C. Analysis of the other segments gives insight into the properties of the contacts and junctions.

In fact narrower ribbons do not make a big difference in the contrast between the edge state and the bulk state contributions to the transport. While it is true that the bulk is less important for narrower ribbons, we have explicitly shown here that, in the wide ribbons measured here and at the charge neutrality point, the bulk actually contributes at most 10 % and as little as 3 % to the conductance (Fig. 3 a, e). Further note that the conductance contributions from the junctions are more important than the bulk contribution and that they require the attention that we gave to them.

In summary, based on these results we can confidently claim that the edge state exists for ribbons with widths at least from 40 nm to 800 nm, in the samples mentioned above similarly to that observed in the sidewall ribbons. Besides, the careful analysis revealed a host of other unexpected novel properties that we also present, beyond the simple fact that the edge state conductance is $1 G_0$, in particular the absence of a Hall voltage.

We have included measurements for different widths and lengths mentioned above (SI16).

(2) “Missing control data. The paper states “We find that only thermally annealed epigraphene structures have an edge state.⁵⁹” The paper needs to include control data of nanoribbons with same dimensions, comparing with and without the thermal annealing, that is believed to repair the edge-mode transport.”

In fact we have made control measurements the Referee refers to, as we explained in response to comment 1. In SI16 we now show 4-point measurements of RIE produced ribbons before and after annealing, which indicates that the annealing is required to produce the edge state in RIE etched ribbons. It is not required for ICP produced ribbons as we show in the main text.

(3) “The paper makes overly broad claims about literature and field. For example, the paper states, “This makes epigraphene the only technologically viable graphene nanoelectronics platform that has the potential to succeed silicon nanoelectronics.” Firstly, the paper seems to define “graphene nanoelectronics platform” as something very specific related to ballistic transport and edge modes whereas most others in the community likely would have a different picture in mind for the words “graphene nanoelectronics platform” such as any use of semiconducting graphene nanoribbons or arrays of graphene nanoribbons or possibly even semiconducting carbon nanotubes integrated into field effect transistors that have a circuit-level energy-delay product less than Si. This is already possible with carbon nanotubes (see L.-M. Peng @ Peking Univ. or S. Mitra atd Stanford). Secondly, “In contrast lithographically patterned exfoliated graphene invariably has highly disordered graphene edges²⁷⁻²⁹ which effectively makes it unsuited for nanoelectronics.” There are multiple reports of edge-repair approaches (see early work from H. Dai, Stanford). The direct growth of surface-synthesized GNRs on dielectrics is now possible, which to many seems to open viable approach to semiconducting graphene and electronics that is not on SiC.”

We agree with the Referee that we have a specific definition for a platform. In fact our definition corresponds quite closely with the original concepts of graphene nanoelectronics as originally proposed (see Refs. 1, 8, 19 and early work by A. Geim and co-workers) where the edge state plays a key role and that the edge state can be gated. Here we take the reasonable criterion for a viable platform is that it is compatible with standard large scale microelectronics processing methods to produce integrated graphene devices. We however disagree that our claims about the field and literature are overly broad (i.e. unsubstantiated). These issues are addressed next.

The Referee challenges our statement in Sup Mat :“In contrast lithographically patterned exfoliated graphene invariably has highly disordered graphene edges²⁷⁻²⁹ which effectively makes it unsuited for nanoelectronics”. The following is a quote from Ref. 27 (Epping et al. *Insulating State in Low-Disorder Graphene Nanoribbons*, *Physica Status Solidi (b)*, 256 (2019) 1900269) that we use to justify this statement:

“To date, in substrate-supported graphene nanostructures, the disordered potential landscape controls the transport properties.¹⁹⁻²⁵ An alternative way to achieve very high electronic quality is placing graphene on hBN, which can significantly reduce the disorder potential.²⁶⁻³⁰ However, in nanostructures, the contribution of edge disorder to the overall disorder remains significant.^{31, 32}”

This is from a 2019 paper from authoritative authors, that clearly states that edge disorder controls the transport properties on substrate supported graphene nanostructures in general and that no solution has been found. We have not found any paper that disputes this claim. The paper further specifies that supported graphene nanostructures always have a disorder induced mobility gap which, as first shown by Kim et al, in 2009 (Ref. 29), makes them unsuited for electronics. In contrast, annealed epigraphene edges are chemically bonded to the SiC substrate giving them the required mechanical and chemical stability. Furthermore, substrate bonding also protects the edges from spontaneous (Peierls) distortions and edge reconstructions that can produce an insulating state even in perfect nanoribbons.

The referee is confident that future efforts, using edge repair methods might remedy the persistent edge disorder problem. We don't deny that, but it has not happened yet to our knowledge. It remains to be seen if the recent development of single layer graphene directly grown on insulator (sapphire) at the wafer scale is viable as a nanoelectronics platform and specifically if it resolves the edge disorder problem and if it is compatible with industrial nanopatterning technology. In any case, and to the point, it has not been demonstrated yet. Furthermore after two decades of research, ballistic edge state transport has not been demonstrated in the top-down methods the referee refers to. Hence, we stand by our broad claim that graphene ribbons on a silicon carbide substrate are the first to demonstrate edge state transport that is not controlled by the disordered potential landscape mentioned by Epping et al. Clearly, we are the first to solve this persistent edge disorder problem in conventionally patterned graphene.

If one insists that a viable graphene nanoelectronics platform must involve semiconducting nanoribbons (i.e. with a bona fide band gap, not a mobility gap) then currently no such platform exists, despite the extraordinary atomic edge precision

reached in bottom-up nanoribbon growth on metal (same problem of upscaling, transfer, placement and contacts as carbon nanotubes, see for instance Adv Mater, 32, 200893 (2020), JACS 144, 11499(2022)). Moreover even if a graphene-based platform using conventional field effect transistors is realized, it is debatable if it can replace Si. The general consensus in the electronics community has always been that a successor for silicon-based CMOS electronics ultimately must be found for high performance nanoelectronics. This implies a *paradigm shift* involving not only in the material but also in the basic operating principles (i.e. spintronics, plasmonics, quantum coherent electronics, etc. see Ref. 7). The graphene edge state electronics that we propose here is consistent with such a paradigm shift and we have demonstrated that the SiC platform is a viable starting point for the reasons we give in the paper.

In conclusion, we appreciate that the Referee does not refute that we have demonstrated excellent and novel properties of the zero-energy edge state, or that we have a viable platform for a novel electronics technology that could challenge Si. The Referee effectively states that a conventional graphene transistor based nanoelectronics platform might be developed in the future once the edge state disorder problem is solved for exfoliated graphene. It should also be mentioned that beside producing a high performance switch, there remains a host of unsolved large-scale nano-fabrication problems that plague not only exfoliated graphene but all bottom-up approaches including nanotube electronics as we mention (see Ref. 6, 10). These persistent problems are resolved in epigraphene as we explain in the text. Consequently, we believe that the title of the paper, that is clearly explained in the abstract, is justified and we hope to have provided compelling arguments here.

(4) “The protected edge state has a mean free path that is greater than 50 microns” at what temperature? Please specify temperature of data presented in each figure.

We thank the referee for this question and we added temperatures to the figures captions. Temperature dependent measurements of a zigzag segments E and H in series (Fig. S13c) shows that the conductance at CNP is $1 G_0$ within a few percent from 2K to 150 K. The $>50 \mu\text{m}$ mfp is accurately determined at $T=4$ K. Note that at $T=300$ K, the conductance slightly *increases* to about $1.1 G_0$. This is most likely caused by thermal broadening of the bulk states which enhances their (small) contribution at CNP. This interpretation is supported by the fact that this increase is not seen in narrower sidewall ribbons (see Fig. S19c). We therefore tentatively conclude that, as for sidewall ribbons, the conductance of the edge state does not significantly change with increasing temperature.

In contrast to sidewall ribbons, the high temperature measurements we refer to are made in a magnetic field to quench the excited edge state as described in the text. As shown in Fig. S13 a without a magnetic field, the conductance *increases* to about $2G_0$ at room temperature due to the proposed excited state (observed in sidewall ribbons) that apparently is quenched in a magnetic field.

The energy of the excited state in sidewall ribbons has been experimentally found to be on the order of 100 meV, see Ref. 41. We estimate this state be about 10 meV in 800 nm wide ribbons (see text).

In any case, more work needs to be done to precisely characterize properties at higher temperatures, but there is already strong evidence that the temperature dependence is weak so that the exceptionally large mfp's most likely persist up to room temperature and in particular on the properties of the excited state.

Note that it is clear that the energy of the excited state can be tuned by the width in so that it can be used in novel electronic devices structures.

REVIEWER COMMENTS

Reviewer #1 (Remarks to the Author):

The authors have addressed my questions adequately. I have no further comments or questions. I would suggest to accept this manuscript for publication.

Reviewer #2 (Remarks to the Author):

1) The inclusion of the data in Fig. S20 for narrower segments of width of about 90 nm is helpful in that now the reader can compare the characteristics of segments of different widths (specifically 90 nm wide in Fig. S20 to about 1000 nm wide in Fig. 3). The point of the previous comment about needing to measure nanoribbons of various widths especially narrow ones was that in the manuscript's Eq. 1, the bulk contribution has a width dependence, but the edge contribution does not. Thus, if the bulk "contributes at most 10 % and as little as 3 % to the conductance" as the authors indicate in their response letter, then there should be little to no width dependence to the segment resistance at fixed L (since transport will be dominated by the edge mode and because a narrow and wide segment have the same number of edges). However, for the $W = 90$ nm segment, the resistance at a L of about 3 μm is close to 85k Ω (from Fig. S20d) whereas for the $W = 1000$ nm S2 sample segment B, the resistance at a L of about 3 μm is only 15.5 k Ω .

Why is there a width dependence in light of the above? Edge transport should result in a graph of resistance versus width (at a given length) that is flat. The data above in which resistance changes with width is more reminiscent of bulk transport.

2) "This makes epigraphene the only technologically viable graphene nanoelectronics platform that has the potential to succeed silicon nanoelectronics."

This statement contains within it unnecessary speculation about what will or will not be possible using other methods to grow graphene nanomaterials, nanostructures. Please remove from abstract. It is inaccurate to say that graphene nanoribbons grown via on-surface polymerization or graphene nanoribbons grown in h-BN trenches or by other methods have no potential. A large fraction of the field would assert there is potential.

arXiv:2208.03145v1

Wang, H.S., Chen, L., Elibol, K. et al. Towards chirality control of graphene nanoribbons embedded in hexagonal boron nitride. *Nat. Mater.* 20, 202–207 (2021). <https://doi.org/10.1038/s41563-020-00806-2>
Wang, H., Wang, H.S., Ma, C. et al. Graphene nanoribbons for quantum electronics. *Nat Rev Phys* 3, 791–802 (2021). <https://doi.org/10.1038/s42254-021-00370-x>

3) "Main

The high electronic mobility and long mean free paths in rolled up graphene sheets (i.e. carbon nanotubes) suggested a graphene nanoelectronics platform as a successor to the silicon nanoelectronics platform. 1-7"

The paper must define "graphene nanoelectronics platform" at the very beginning since the authors have a very specific meaning behind "graphene nanoelectronics platform" that differs from what the words only would suggest. Just providing references to their definition or explanation in the SI is not enough.

Response to Reviewer 1

We thank Reviewer 1 for his/her positive response and appreciation of the paper.

Response to Reviewer 2.

(1) The inclusion of the data in Fig. S20 for narrower segments of width of about 90 nm is helpful in that now the reader can compare the characteristics of segments of different widths (specifically 90 nm wide in Fig. S20 to about 1000 nm wide in Fig. 3). The point of the previous comment about needing to measure nanoribbons of various widths especially narrow ones was that in the manuscript's Eq. 1, the bulk contribution has a width dependence, but the edge contribution does not. Thus, if the bulk "contributes at most 10 % and as little as 3 % to the conductance" as the authors indicate in their response letter, then there should be little to no width dependence to the segment resistance at fixed L (since transport will be dominated by the edge mode and because a narrow and wide segment have the same number of edges). However, for the $W = 90$ nm segment, the resistance at a L of about 3 μm is close to 85 kohm (from Fig. S20d) whereas for the $W = 1000$ nm S2 sample segment B, the resistance at a L of about 3 μm is only 15.5 kohm.

Why is there a width dependence in light of the above? Edge transport should result in a graph of resistance versus width (at a given length) that is flat. The data above in which resistance changes with width is more reminiscent of bulk transport.

Response to (1)

The ribbons in the main text and in Fig. S20d cannot be compared in the way that the reviewer attempts to do it because they differ not only in their width but also in their bulk mobilities, their edge state mean free paths and their charge densities that all contribute to the total conductance. However they both demonstrate the identical single channel, $1G_0$, ballistic transport.

Explicitly, as is clear from Eq. 1, the conductance of a ribbon depends on its length L , its width W , its bulk mobility μ and the edge state mean free path λ and charge density n . The edge mean free path λ and the bulk mobility μ are material properties that depend on how the wafer was processed, which SiC crystal face is used, and also significantly, along which crystallographic axis the ribbons are aligned (i.e. armchair or zigzag or something in between). The charge density n can be varied with a top gate so that at charge neutrality $G = G_0/(1 + \lambda/L)$. The bulk conductivity is $\sigma = ne\mu$, where e is the electronic charge, so that the bulk conductance is $G_{\text{bulk}} = ne\mu W/L$. The edge state conductance is $G_{\text{edge}} = G_0/(1 + \lambda/L)$, so that the total conductance of a ribbon (in a 4-point measurement) is

$$G = G_0/(1 + \lambda/L) + ne\mu W/L. \quad [1]$$

$$\text{and the resistance is } R = 1/\{ G_0/(1 + \lambda/L) + ne\mu W/L \}. \quad [2]$$

The ribbons in Fig. S20d are not well aligned along the zigzag direction, they do not have a top gate and are produced on the 4H-SiC (0001), so that they are charged. Ballistic transport is determined from the (nonlinear) dependence of R on L in the same ribbons (i.e. at fixed W , λ , μ and n). The $1G_0$ conductance is found by fitting the measured $R(L)$ to equation 2 and for Fig. S20d, this gives $\lambda = 0.46\mu\text{m}$, $0.94\mu\text{m}$ and $1.2\mu\text{m}$ before and after annealing, and measured in

vacuum, respectively. The bulk mobility is found to be $\mu=250 \text{ cm}^2 \text{ V}^{-1} \text{ s}^{-1}$, which corresponds to a bulk mean free path $\lambda_{\text{bulk}} = 4 \text{ nm}$, and the charge density $n= 5 \times 10^{12} \text{ cm}^{-2}$ (for further details, see Yue Hu, thesis, chapter 3). Hence the edge state mean free path is 300 times greater than the bulk mean free path, but it is not as large as we found for the zigzag ribbon S2.

The bulk mobility of S2 (which is a zigzag ribbon) is $\mu= 700 \text{ cm}^2 \text{ V}^{-1} \text{ s}^{-1}$ and the edge mean free path is $\lambda=35 \text{ }\mu\text{m}$, while for the armchair ribbon sample S1, $\lambda=6 \text{ }\mu\text{m}$.

Hence the variability the referee has pointed out are all due to the way the ribbon is produced (which we continue to perfect). Regardless, in all cases we find that the conductance follows Eq. 1 and there is no inconsistency.

In order to make these points more clearly, we have included a reference to Yue Hu's thesis that explains Fig. S20d in more detail.

(2) *“This makes epigraphene the only technologically viable graphene nanoelectronics platform that has the potential to succeed silicon nanoelectronics.”*

This statement contains within it unnecessary speculation about what will or will not be possible using other methods to grow graphene nanomaterials, nanostructures. Please remove from abstract. It is inaccurate to say that graphene nanoribbons grown via on-surface polymerization or graphene nanoribbons grown in h-BN trenches or by other methods have no potential. A large fraction of the field would assert there is potential.

arXiv:2208.03145v1

Wang, H.S., Chen, L., Elibol, K. et al. Towards chirality control of graphene nanoribbons embedded in hexagonal boron nitride. Nat. Mater. 20, 202–207 (2021).

<https://doi.org/10.1038/s41563-020-00806-2>

Wang, H., Wang, H.S., Ma, C. et al. Graphene nanoribbons for quantum electronics. Nat Rev Phys 3, 791–802 (2021). <https://doi.org/10.1038/s42254-021-00370-x>

Response to (2)

We thank the referee for pointing this out. We certainly do not want to imply any prediction for what may happen in the future, and we do not want to speculate on the potential of on-surface polymerization or graphene grown in h-BN trenches so we modified the statement in the abstract to read:

“This makes epigraphene currently the only technologically viable graphene nanoelectronics platform that has the potential to succeed silicon nanoelectronics.”

Which we further justify below.

Schwierz's authoritative paper, "Graphene Transistors" in Nature Nanotechnology 5, pages 487–496 (2010) (4000+ citations) states:

"At present, the most popular approaches to graphene preparation are mechanical exfoliation¹, growth on metals and subsequent graphene transfer to insulating substrates^{20,21}, and thermal decomposition of SiC to produce so-called epitaxial graphene on top of SiC wafers^{22,23}. Exfoliation is still popular for laboratory use but it is not suited to the electronics industry, whereas the other two options both have the potential for producing wafer-scale graphene. After the graphene has been prepared, common semiconductor processing techniques (such as lithography, metallization and etching) can be applied to fabricate graphene transistors."

The examples that the referee supplies to make his/her point (which we assume are representative of the state-of-the-art top-down processes) involve devices, methods and properties that fall short of those required in a viable electronics platform. Specifically:

1. The nanoparticle-catalysed lithography that is used in Wang et al [Towards chirality control of graphene nanoribbons embedded in hexagonal boron nitride. Nature Materials 20, 202 (2021)] are not common semiconductor processing techniques, and as presented it does really qualify as a lithography method since the trenches are at random locations.
2. The transport properties are not useful for conventional FET electronics, since they do not have a band gap at room temperatures and no alternative form of electronics is mentioned.
3. There is no indication what strategy will be used for upscaling and for reliable nanowire interconnections as required in conventional commercial electronics; currently metal contacts to these devices are fragile had have high resistances and vast numbers (many millions if not billions per cm²) will be required for electronics, which by the estimation of O. Braun et al [Optimized graphene electrodes for contacting graphene nanoribbons, Carbon, 184 331 (2021)] still poses daunting technical challenges.
4. The substrates are not conventional: the devices are produced on exfoliated micrometer size BN flakes that are produced by two of the coauthors, who are still so far of only 3 producers of these non-commercial flakes in the world.
5. No integration strategy with Si based electronics is mentioned nor is it clear if this work is attempting to succeed high performance Si electronics and how that will be done, which is ultimately the goal of post-CMOS electronics.

We also should point out that the data appear to be more consistent with a disorder induced transport gap rather than a band gap similar to those demonstrated by M. Y. Han, J. C. Brant, and Philip Kim Phys. Rev. Lett. 104, 056801 (2010), which does not qualify as viable for electronics at this time. Moreover, the arXiv:2208.03145v1 paper that is also cited by Reviewer 2 actually only shows that bottom-up produced ribbons can be metallic (with a conductance $\approx 1/10 G_0$). We don't think that these data are sufficient to claim quantum (i.e. quantized ballistic) transport that requires the conductance to be a multiple of $1G_0$. In summary, although these physical properties are interesting but a clear route to a viable technology [C. R. Eddy. & D. K. Gaskill, Silicon Carbide as a Platform for Power Electronics. Science 324, 1398 (2009)] certainly cannot be claimed.

(3) The high electronic mobility and long mean free paths in rolled up graphene sheets (i.e. carbon nanotubes) suggested a graphene nanoelectronics platform as a successor to the silicon nanoelectronics platform.1-7”

The paper must define “graphene nanoelectronics platform” at the very beginning since the authors have a very specific meaning behind “graphene nanoelectronics platform” that differs from what the words only would suggest. Just providing references to their definition or explanation in the SI is not enough.

Response to (3)

We thank the referee for this suggestion and we included the following paragraph at the very beginning which clarifies the meaning of a viable graphene nanoelectronics platform.

Main

Currently, silicon provides a platform for high performance complementary metal oxide semiconductor (CMOS) nanoelectronics that enables billions of transistors to be patterned on an area of 1 cm² using the most highly developed technologies in the world. However, the CMOS platform is reaching its scaling limits [1]. A viable continuation of electronics technology will require a new approach to electronics, while preserving as much as possible current industrial fabrication methods [2]. Hence, this new platform should involve a single crystal substrate and a conventionally nanopatterable material. We show here that epigraphene is perfectly suited. In fact SiC already is a platform for high power electronics [3] so that it is fully compatible with microelectronics processing methods.

[1] A. Chen, J. Hutchby, V. Zhirnov, G. Bourianoff, Emerging Nanoelectronic Devices, Wiley publisher (2014).

[2] F. Schwierz, “Graphene Transistors” in Nature Nanotechnology 5, 487–496 (2010).

[3] C. R. Eddy Jr. and D. K. Gaskill, Silicon Carbide as a Platform for Power Electronics, Science 324, 1398-1400, (2000).

REVIEWER COMMENTS

Reviewer #2 (Remarks to the Author):

(A) Follow up on previous point 1.

The edge mfp in one case is 1 micrometer and in another it is 35 micrometers.

The authors make the argument that “The variability the referee has pointed out are all due to the way the ribbon is produced (which we continue to perfect).”

This explanation could very well be. OR, the assumptions being made when modeling the bulk versus edge mfp are wrong in an unexpected way. The authors have a nice story but this discrepancy could be a hole in the story. Especially since the discrepancy is going in the direction of overall lower conductance with decreasing nanoribbon width. This casts some doubt in my mind.

I am willing to give my approval that this is published if the main text of the paper clearly makes note of the possible large difference in edge mfp between samples and indicates the authors’ hypothesis that the differences are in the way the ribbon is produced. This way the reader can be more aware and better informed. This will help the community either build from this work in the future (by having better knowledge that some unknown factor is causing variability so that efforts can be made to eliminated it) or more quickly identify the hole in the models.

(B) Follow up on previous point 2.

“This makes epigraphene currently the only technologically viable graphene nanoelectronics platform that has the potential to succeed silicon nanoelectronics.”

Currently, epigraphene has significant challenges that must be worked through (e.g., need for very low temperature, large variations in edge mfp, need for demonstration of even a basic functional circuit), just like other graphene nanoribbon approaches have challenges that must be worked through (that the authors list). No approaches are currently ready. This statement still needs revision.

(C) Follow up on previous point 3.

The new 1st paragraph still does not explicitly define what “graphene nanoelectronics platform.” A sentence such as “We define a graphene nanoelectronics platform as”. Or “This paper defines a graphene nanoelectronics platform as...” followed by the critical aspects of a graphene nanoelectronics platform as envisioned by the authors is needed. The readers will be confused without a clear definition!

Answer to Reviewer #2's remarks to the Authors

Reviewer 2's comments are in bold face type.

(A) Follow up on previous point 1.

The edge mfp in one case is 1 micrometer and in another it is 35 micrometers.

Response

That is correct. In case 1 we are dealing with an armchair ribbon and in case 2 we are dealing with a zigzag ribbon. This difference is thoroughly discussed in the paper.

The authors make the argument that “The variability the referee has pointed out are all due to the way the ribbon is produced (which we continue to perfect).” This explanation could very well be [true].

Response

As extensively explained in the paper, and consistent with theory, the difference in the mean free path is primarily due to the difference in transport in armchair and zigzag ribbons, and this has been noted in earlier papers as well. Reiterating, theory predicts that armchair edges are generally insulating and zigzag edges are metallic. Theory further predicts that (chiral) ribbons oriented in between the two are generally metallic, but we find experimentally that their metallicity (i.e. mean free path) is reduced. Furthermore, theory predicts that the edge state is sensitive to edge defects, and hence expects that the fewer edge defects the longer the mean free path. So if we want larger mean free paths, we have to cut the ribbons along the zigzag direction and we have to minimize defects. Better annealed ribbons have better edges. Hence, like in crystal growth in general, the production process has to be perfected in order to get more perfect crystals. All of these points have been made in the text.

OR, the assumptions being made when modeling the bulk versus edge mfp are wrong in an unexpected way.

Response

Apparently, the Reviewer casts doubt on the “modeling” of the bulk and the edge state that could be wrong in an “unexpected” way. We are not sure what assumptions Reviewer 2 is referring to. The theories and methods used here are standard and have been well accepted in the mesoscopic community for over half a century. We use conventional mesoscopic transport theory without any modification (see for example chapter 2 in Ref. 53: S. Datta, *Electronic transport in mesoscopic systems*, Cambridge University Press, Cambridge, 1995). Explicitly, Equation 1 in the paper is the Landauer equation, which is a universally used general equation to describe transport in mesoscopic systems. It has been ubiquitously used to describe graphene transport for the past 20 years. It applies equally to ballistic states, eg edge states, and diffusive states, eg bulk states; the only

parameter is the mean free path that therefore can be unambiguously determined experimentally as we have done. Moreover we used the same theory to characterize the epigraphene edge state in Ref. 30: J. Baringhaus et al. *Exceptional ballistic transport in epitaxial graphene nanoribbons*, Nature, 506 (2014) 349-354. The Reviewers of that paper expressed no doubt about these basic concepts. There is no doubt that the “modeling” is fundamentally correct and there is no reason to expect that it is wrong in some unexpected way that could substantially change the conclusions. In addition, we went to great lengths to experimentally verify Eq. 1 to such an extent that the Reviewer previously characterized it as tedious.

The authors have a nice story but this discrepancy could be a hole in the story. Especially since the discrepancy is going in the direction of overall lower conductance with decreasing nanoribbon width. This casts some doubt in my mind.

Response

We are pleased that the Reviewer likes our story. However apparently, the Reviewer is convinced that there is a discrepancy where there is in fact none. Our earlier work in 2014, (Ref. 30) shows that zigzag ribbons that are 40 nm wide have mean free paths that vary from 4 μm to 58 μm . These variations are due to the quality of the edge as we have determined in the 8 years since then. In the range of widths that we have examined, we have not found any correlation between mean free path and width. In fact the mean free path is not theoretically expected to decrease with decreasing width, and arguments can be made for the opposite effect. We have in fact been looking for that! More precisely, since the edge state is a coherent effect involving both edges, as we explicitly show in the paper, we may expect that when the width is larger than the coherence length, the edge state should disappear! We discuss this point in the paper. I think that the Reviewer is confused because it is well known that the mean free paths of narrow ribbons produced from exfoliated graphene do in fact decrease with decreasing width due to disorder caused by the lithography. We find that this is not so for annealed epigraphene ribbons in general. We have explained and justified this effect in terms of bonding of the edges to the SiC substrate which we justify citing appropriate theory.

I am willing to give my approval that this is published if the main text of the paper clearly makes note of the possible large difference in edge mfp between samples and indicates the authors' hypothesis that the differences are in the way the ribbon is produced. This way the reader can be more aware and better informed.

Response.

We appreciate Reviewer 2's positive recommendation for publication, but again, we are confused by the comments. We do clearly make note of the difference between the mean free paths. The main difference is between zigzag and armchair ribbons. In fact that is

the central point of Fig.3. Moreover, as we point out, the bonding to the substrate plays a fundamental role in stabilizing the edges and armchair “defects” cause edge state scattering. It is experimentally clear that the more these defects are eliminated the larger the mean free path.

This will help the community either build from this work in the future (by having better knowledge that some unknown factor is causing variability so that efforts can be made to eliminated it)

Response

The paper is written such that the experiments can be easily reproduced and there is no major unknown factor that we know of. As for crystal growth in general, progress involves eliminating defects, which typically involves optimizing production methods (i.e. annealing times and temperatures and careful alignment of the sample to select zigzag directions.)

... or more quickly identify the hole in the models.

Response

We are surprised by Reviewer 2’s insistence that there must be a hole in the model that we are using. Landauer’s theory is so basic and so well accepted that a hole in it would imply a hole in fundamental concepts of mesoscopic transport.

(B) Follow up on previous point 2.

“This makes epigraphene currently the only technologically viable graphene nanoelectronics platform that has the potential to succeed silicon nanoelectronics.”

Currently, epigraphene has significant challenges that must be worked through (e.g., need for very low temperature, large variations in edge mfp, need for demonstration of even a basic functional circuit), just like other graphene nanoribbon approaches have challenges that must be worked through (that the authors list). No approaches are currently ready. This statement still needs revision.

Response

While we disagree with the Reviewer on several points here (i.e. low temperatures are not needed since the edge state exists at room temperatures (Fig. S20, and Fig. S19 from ref [30]) and for all practical purposes, the mean free path needs not be much larger than a few microns), we can in fact in principle produce interconnected edge state circuits as dense as is realized in conventional silicon nanoelectronics. We know in principle how to make an edge state switch but it is true that we still need to demonstrated it. So, not to

distract from the primary message of the paper, and to satisfy the Reviewer, we made the following modification.

“This makes epigraphene currently the only technologically viable graphene nanoelectronics platform that has the potential to succeed silicon nanoelectronics.”

Changed to

“This makes epigraphene a technologically viable graphene nanoelectronics platform that has the potential to succeed silicon nanoelectronics.”

And to be consistent we changed the title to

“An epitaxial graphene platform for zero-energy edge state nanoelectronics”

We also added in the introduction that the edge state is observed at room temperature:

“edge state transport was first observed in 2014 up to room temperature in self-assembled 40 nm wide graphene ribbons grown on the sidewalls” and “Exceptional edge state transport is observed with unprecedented mean free paths exceeding 50 microns even at room temperature.”

(C) Follow up on previous point 3.

The new 1st paragraph still does not explicitly define what “graphene nanoelectronics platform.” A sentence such as “We define a graphene nanoelectronics platform as”. Or “This paper defines a graphene nanoelectronics platform as...” followed by the critical aspects of a graphene nanoelectronics platform as envisioned by the authors is needed. The readers will be confused without a clear definition!

Response

We now start the paper with:

“A viable nanoelectronics platform can be defined as a material that can be processed using conventional nanoelectronics technology as is required to produce high density, high performance commercial nanoelectronics.”

Along these lines, we slightly modified the first paragraph of the introduction to be more explicit what we mean by viable continuation of electronics.

Note that due to editorial constrains, we have shortened the abstract; we have also split Fig 3 & 4 for better readability.